# In silico cancer immunotherapy trials uncover the consequences of therapy-specific response patterns for clinical trial design and outcome

Jeroen H. A. Creemers [1,2], Ankur Ankan[3], Kit C. B. Roes[4], Gijs Schröder [3], Niven Mehra [5], Carl G. Figdor [1,2], I. Jolanda M. de Vries [1] & Johannes Textor [1,3] ✉

Late-stage cancer immunotherapy trials often lead to unusual survival curve shapes, like delayed curve separation or a plateauing curve in the treatment arm. It is critical for trial success to anticipate such effects in advance and adjust the design accordingly. Here, we use in silico cancer immunotherapy trials – simulated trials based on three different mathematical models – to assemble virtual patient cohorts undergoing late-stage immunotherapy, chemotherapy, or combination therapies. We find that all three simulation models predict the distinctive survival curve shapes commonly associated with immunotherapies. Considering four aspects of clinical trial design – sample size, endpoint, randomization rate, and interim analyses – we demonstrate how, by simulating various possible scenarios, the robustness of trial design choices can be scrutinized, and possible pitfalls can be identified in advance. We provide readily usable, web-based implementations of our three trial simulation models to facilitate their use by biomedical researchers, doctors, and trialists.

Immunotherapy is revolutionizing the treatment landscape for patients with advanced cancers. While the number of immuno-oncology drugs under investigation is rising rapidly – around 4700 agents are currently in the development pipeline – the need to further improve patient outcomes remains high[1]. Well-designed immunotherapy trials are crucial to establish advances in clinical outcomes robustly. Unfortunately, the odds for cancer treatments to successfully pass the development pipeline are unfavorable, and only a minority of the treatments (5–10%) ultimately obtain market approval[2–4]. Even for cancer therapies that do reach late-stage development, approval rates remain modest at around 27%[5]. The primary reason in most of these

trials (i.e., 63.7%) is failure to demonstrate efficacy[5], which can be partly attributed to suboptimal trial design choices based on overly optimistic assumptions of the treatment effect. Such assumptions may be used to erroneously justify low numbers of patients or inappropriate endpoints and lower the power of these trials[5,6].

Immunotherapy trials raise complex design questions, and conventional design methods are not always a good match to the unique characteristics of immunotherapies[7]. There is a very broad spectrum of therapies based on various molecular mechanisms – ranging from immunomodulators to cell therapies, cancer vaccines, oncolytic viruses, and CD3-targeted bispecific antibodies – that can lead to unusual

[1]Medical BioSciences, Radboud university medical center, Nijmegen, The Netherlands. [2]Oncode Institute, Nijmegen, The Netherlands. [3]Data Science group, Institute for Computing and Information Sciences, Radboud University, Nijmegen, The Netherlands. [4]Department of Health Evidence, Section Biostatistics, Radboud university medical center, Nijmegen, The Netherlands. [5]Department of Medical Oncology, Radboud university medical center, Nijmegen, The Netherlands. ✉e-mail: johannes.textor@ru.nl

toxicity profiles, response patterns, and survival kinetics[8–10]. These observations render a "one-design-fits-all" approach futile and stress the need for designs that are tailored to immunotherapy or even combination therapies.

Immunotherapies are known to induce a delayed clinical effect and long-term overall survival (OS) in only a subset of patients[11]. The survival curve reflects these phenomena by a delayed curve separation and a plateau of the treatment arm at later stages of the trial[12]. These characteristics violate a fundamental premise that underlies the design of many trials: the proportional hazard assumption (PHA) – essentially stating that the treatment effect should remain constant over time[13]. As a result, immunotherapy trials based on this principle can have an overestimated power[12,13] and require a longer follow-up to demonstrate efficacy than initially planned[12], increasing the likelihood of a negative trial.

These issues led to the development of innovative methods such as novel radiological criteria to quantify tumor responses[9,14,15], (surrogate) endpoints to capture unique survival kinetics[10,16–19], biomarkers to enrich for patients more likely to respond to treatment[20–23], and statistical methods to retain a trial's power in the presence of unusual survival kinetics[24–26]. Despite the multitude of available methods, it is difficult to predict trial outcomes in advance and select the methodology accordingly. The stakes are high: a trial design built on accurate predictions of the response kinetics is more likely to be positive, whereas misjudgment could result in a negative trial, potentially compromising patient benefit, vast amounts of work, and (public) research funds.

In this work, we use late-stage in silico cancer immunotherapy trials to investigate how design decisions affect the trial outcome in the context of cancer immunotherapy, possibly combined with chemotherapy. The mechanism-based nature of these trials allows researchers to translate cellular processes in the tumor microenvironment and immunotherapeutic interventions thereon into predicted response patterns, survival kinetics, and trial outcomes. An in silico immunotherapy trial is based on explicit biological assumptions and provides an intuitive means to predict risk profiles and treatment efficacy. Moreover, it equips researchers with a tool to scrutinize trial designs and analysis strategies of upcoming trials in advance to identify potential risks and pitfalls. We use three different simulation models to perform our in silico trials, based on work by ourselves[27] and other authors[28,29]. Despite considerable differences, all models replicate late-stage immunotherapy or combination trials reasonably well and capture their typical survival kinetics. Then, we demonstrate various applications of such trial simulations, including the ability to scrutinize a clinical trial's design and sample size calculations based on a range of predicted possible outcomes. Finally, we illustrate the consequences of (not) considering immunotherapy-specific response patterns in settings selected for educational purposes, such as selecting survival endpoints and randomization ratios of upcoming trials and planning interim analyses.

## Results

### Generating trial populations based on tumor-immune dynamics

We used in silico cancer immunotherapy trials based on mechanistic simulations of cancer-immune dynamics to investigate the consequences of immunotherapy-specific response patterns on trial design principles[26]. The virtual patients in these trials are simulated with ordinary differential equation (ODE) models, which describe disease courses based on assumptions about interactions between tumor cells and the immune system[26]. In this paper, we will focus on simulating two years of follow-up after treatment – while it is straightforward to consider longer follow-up times with in silico trials, a two-year time frame is common for contemporary immunotherapy trials[30–32].

To investigate the extent to which our simulation results depend on specific modeling choices, we use three different ODE models.

Model 1 (M1) describes the following tumor-immune dynamics in the tumor microenvironment: immunogenic tumor growth leading to priming and clonal expansion of naïve T cells, migration of effector T cells to the tumor microenvironment, and formation of tumor-immune complexes to enable tumor cell killing (see Methods; Fig. 1A). We simulate treating these patients with immune checkpoint inhibitors (ICI), chemotherapy, or both. ICI increase the T cell killing rate and directly affect the tumor-immune dynamics. Chemotherapy has a cytotoxic effect on the tumor, slowing its growth. A detailed description of a previous version of this model, including the rationale for parameter selection, has been published previously[26]; a full description of the version used this paper is given in the Methods. In contrast, model 2 (M2) does not represent T cells migration between lymph nodes and tumor microenvironment; however, it does contain an explicit representation of antigen-presenting cells (APCs)[33]. Finally, Model 3 (M3) does not contain either T cell migration or APCs, but it does take T cell exhaustion into account. Another important difference between the models lies in how tumor growth is represented: M1 uses a size-dependent growth rate, M3 a resource-constrained growth rate (logistic growth), and M2 uses unlimited exponential growth.

Regardless of model specifics, in silico clinical trials describe cancer outcomes on three levels: (1) a cellular level, (2) a patient level, and (3) a trial population level. Cellular interactions in the tumor microenvironment are translated into clinical trial outcomes as follows: firstly, the ODE model is implemented, and model parameters that vary between patients are selected by fitting to existing survival data (Fig. 1B; see Methods). Next, individualized disease trajectories – either treated or untreated – of cancer patients are generated (Fig. 1C). Eventually, patients are randomized into two cohorts to resemble conventional phase III trials: a control group (either placebo or chemotherapy) and a treatment group (immunotherapy, chemoimmunotherapy, or induction chemotherapy followed by immunotherapy; Fig. 1D). Since the cellular dynamics (e.g., tumor burden over time or the efficacy of T cell killing) and survival outcomes of these patients are known and can be modified, in silico clinical trials are suited to answer questions like: "Assuming that a novel treatment increases T cell killing 5-fold, how does this translate to a survival benefit in patients? Moreover, how many patients are needed to establish this benefit in a clinical trial? When should one analyze the results?" (Fig. 1E).

Despite their differing mechanisms, the models generate qualitatively similar predictions (Fig. 2): tumors grow at realistic speeds and are usually not cleared by the immune system without therapeutic intervention. In the models, therapeutic interventions can slow or even reverse tumor growth, in principle leading to two major contrasting outcomes: death or long-term survival. However, there is a unique effect in M3 where even after treatment and growth reverse, the tumor burden keeps oscillating over time, leading to regular self-resolving recurrences. While this may not be entirely realistic, it is not an issue for our purpose as we shall focus on the initial growth trajectory of the tumor preceding and up to 2 years after treatment, and recurrences in M3 happen after that.

### In silico late-stage immunotherapy trials yield realistic survival outcomes

To investigate whether our in silico models can generate realistic survival curves as observed in late-stage immunotherapy trials, we fitted the models to three different datasets: (1) the north central cancer treatment group (NCCTG) lung cancer survival dataset[34]; (2) the CA184-024 trial (ipilimumab+dacarbazine vs. dacarbazine in previously untreated metastatic melanoma[35]); and (3) the CheckMate 066 trial (nivolumab+placebo vs. dacarbazine + placebo in treatment-naive metastatic melanoma patients without BRAF mutation[36]). The choice for these trials is based on the size of the trials and the maturity of the data. The follow-up of the CA184-024 trial and the CheckMate 066 trial were five and three years, respectively. As the last two datasets were

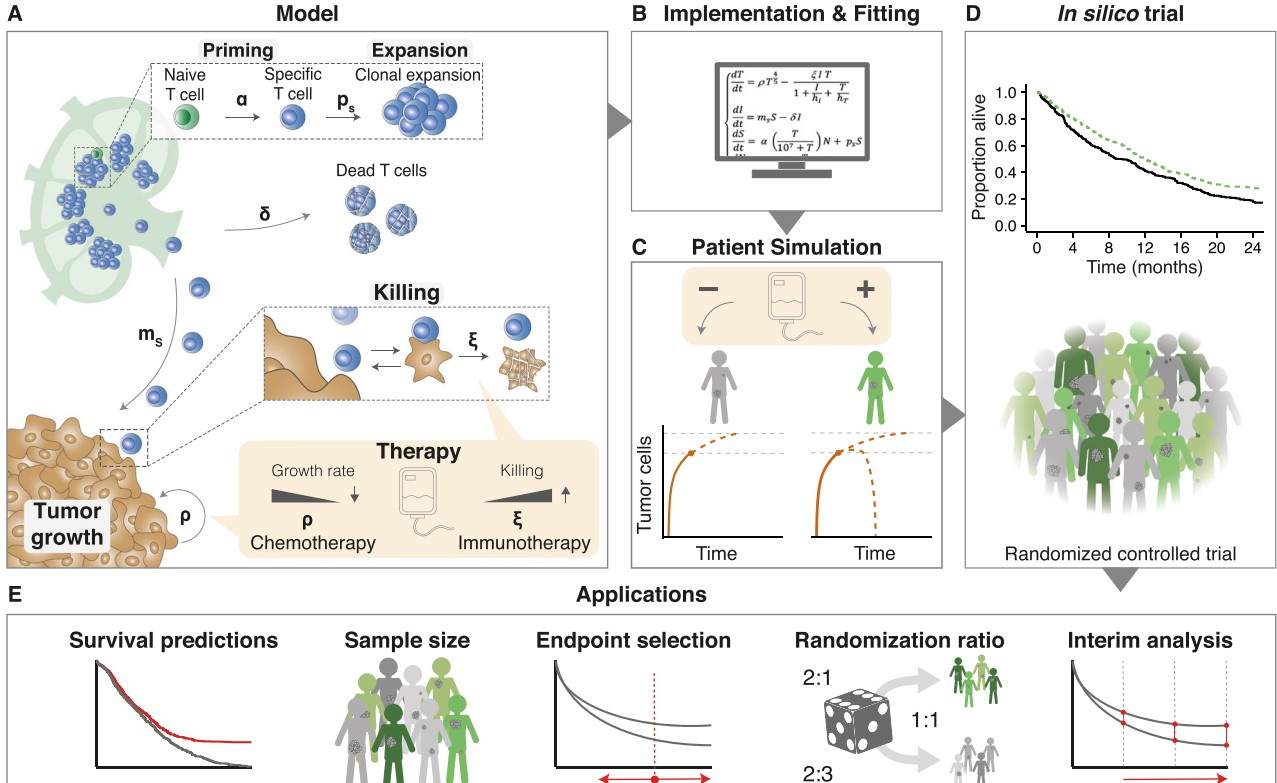

**Fig. 1 | In silico late-stage immunotherapy trials and their applications.**
**A** Cellular interactions between a tumor and the immune system as implemented in ODE model M1 (Methods). This model describes immunogenic tumor growth leading to a T cell response originating from lymph nodes. Disease courses in patients can be steered by immunotherapy, chemotherapy, or a combination of both. Parameters: $\alpha$=naive T cell priming rate, $\delta$=effector T cell death rate, $\xi$=effector T cell killing rate, $\rho$=tumor growth rate, $p_s$=effector T cell proliferation rate, and $m_s$=effector T cell migration rate. **B** After implementation, we used survival data from clinical trials to fit some of the model parameters. **C** Patients received either no treatment (placebo), chemotherapy, immunotherapy, or both. Disease trajectories based on tumor-immune dynamics were simulated for each patient, resulting in individual survival outcomes. **D** Subsequently, cohorts of patients were constructed based on the fitted parameters to simulate actual immunotherapy trials. **E** Applications of such trials include predicting possible survival outcomes of trials, estimating sample sizes needed for a range of scenarios, and investigating endpoints, randomization ratios, and the timing of interim analyses.

not publicly available, we extracted the data using image digitization (see Methods). As a reference for the in silico trials, we visualized the Kaplan–Meier estimators of these datasets (Fig. 3A). Both trials were digitized correctly, as reflected by the nearly identical risk tables compared to the original manuscripts[35,36]. Next, we fitted the tumor growth rate distributions and treatment effect parameters for chemotherapy and immunotherapy (NCCTG: 3 parameters; immunotherapy trials: 4 parameters) to these datasets (M1: Fig. 3B; M2, M3: Supplementary Fig. 1). For the CA184-024 and CheckMate 066 trials, the simulated patients were treated with ICI upon diagnosis, increasing their T-cell killing rates. For simplicity, we did not simulate dropout or censoring in the trials shown in this paper, although it could be added to the simulation. Model M1 achieved satisfactory fits to all datasets. However, M2 and M3 had difficulties fitting the CheckMate 066 data, with M2 predicting more rapid death in the control arm and M3 predicting a cross-over of survival curves. M3 also had difficulties fitting the other two datasets, as its survival curves plateaued from 12 months after treatment onwards. While the fit of all models can be improved by allowing more parameters to vary, we chose to keep the number of fitted parameters small to investigate the consequences of such issues on our downstream analyses.

Hence, our in silico trials couple the disease mechanism and treatment effect to a predicted clinical trial outcome. By allowing model parameters to vary between patients, such models can be fitted to existing clinical trial data. Whether a good fit can be achieved depends on the model assumptions and the number of parameters that are allowed to vary. In our case, models M1 and M2 were able to fit the three datasets reasonably well, with M3 showing a substantially worse fit.

Interestingly, although not incorporated explicitly, the models reproduced hallmark survival curve features arising as a consequence of the interaction between tumor and immune cells typically seen in immunotherapy trials: a delayed curve separation and a plateau of the survival curve of the treatment arm at later stages of the trial (last two columns in Fig. 3B and Supplementary Fig. 1).

## In silico immunotherapy trials predict immunotherapy-specific response patterns

The design and the success rate of any clinical trial depends, among others, on a realistic prediction of the shape of the survival curves and the distribution of clinical outcomes. For late-stage immunotherapy trials, commonly observed immunotherapy-induced response patterns are a delayed curve separation and a plateauing tail of the survival curve of the treatment arm (Fig. 3). These characteristic survival curve shapes violate a vital premise of many clinical trials: the proportional hazard assumption (PHA). The PHA states that the "instantaneous death rate" of a patient (i.e., the hazard rate) in both arms of the trial should be proportional, resulting in a constant hazard ratio. Many traditional design methods, ranging from sample size calculations to outcome analyses, are based on this convenient assumption. For late-stage immunotherapy trials, this induces two problems: (1) while a violation of the PHA

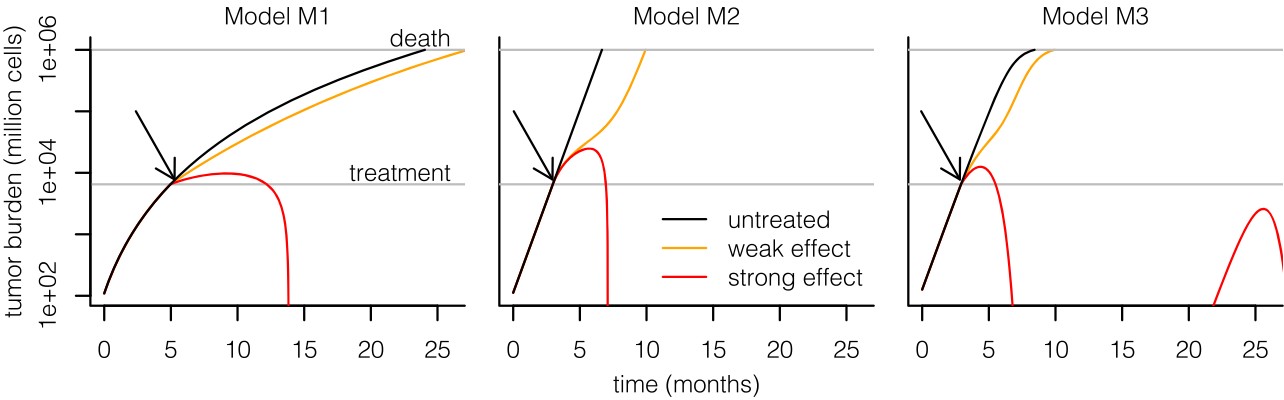

**Fig. 2 | Simulating immunotherapy responses using different mathematical models.** Each simulation starts with a single malignant cell that establishes a tumor. Without treatment, this tumor grows to a lethal volume (upper horizontal line) over the course of several months; the plots start when the tumor has a size of $10^8$ cells. Treatment is started when the tumor reaches a size of $65 \times 10^8$ cells (lower horizontal line). Immunotherapy is implemented in each model by increasing the rate at which T cells kill tumor cells; in M2, the death rate of T cells is additionally decreased by the same factor. The treatment effect sizes are chosen per model such that there is a partial response (orange, leading to prolonged survival) or a complete response (red, leading to tumor eradication). The recurrence of the tumor in M3 is a consequence of the model's equations, which lead to oscillating dynamics of the tumor burden rather than complete eradication in the complete response regime. Arrows indicate start of treatment.

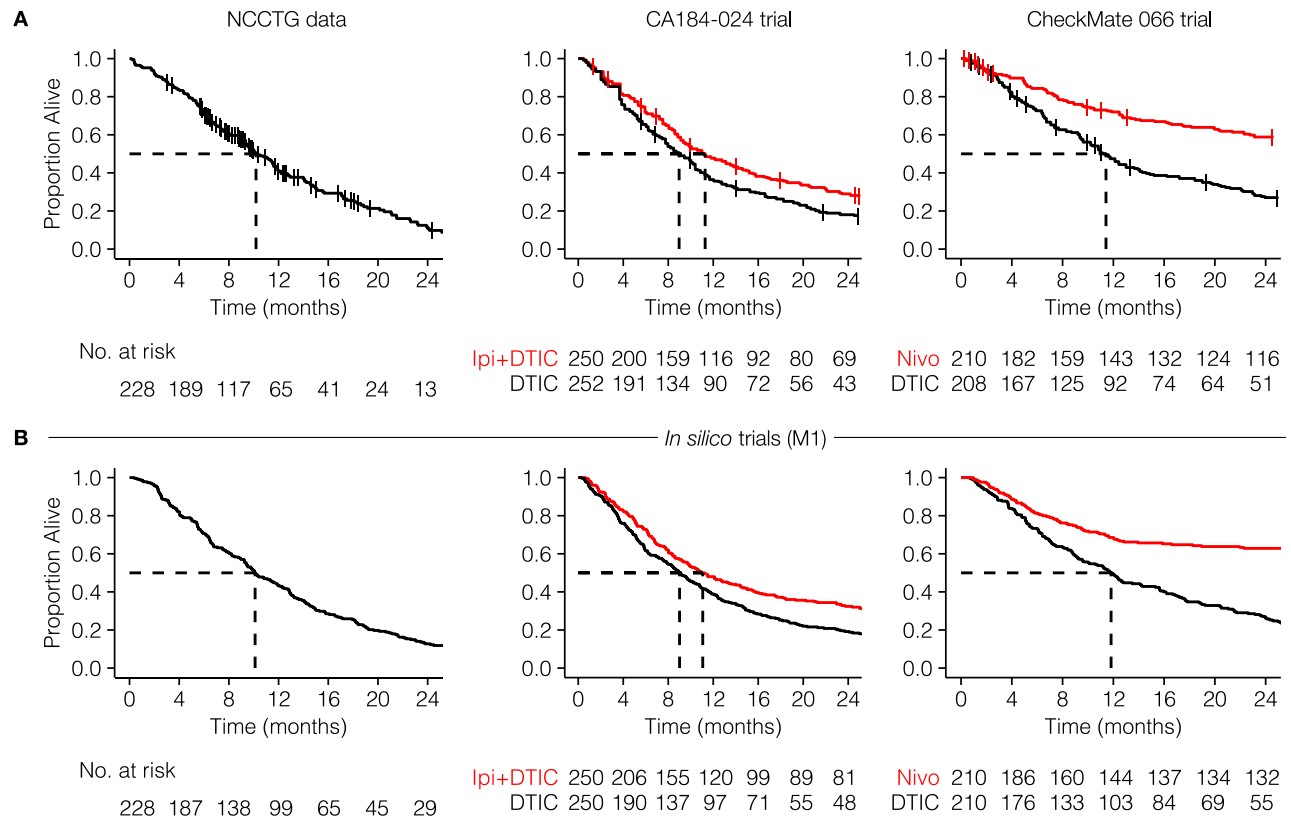

**Fig. 3 | Fitting in silico cancer immunotherapy trial models to survival data.**
**A** Kaplan–Meier estimators of the NCCTG, CA184-024, and CheckMate 066 trials. While the NCCTG dataset is publicly available[34], the others are carefully reconstructed survival curves based on digitized data from the respective articles[35,36]. **B** Trial simulations can generate realistic survival curves as observed in actual immunotherapy trials. Specifically, typical immunotherapy-related survival curve shapes – such as a delayed curve separation and a plateau in the treatment arm – arise from these simulations as emergent behavior. Source data are provided as a Source Data file.

needs to be addressed during trial planning, the hazard rates – and an eventual violation of the PHA – becomes available only after the trial; and (2) if a trial does not adhere to a PHA, what will be the shape of the survival curve? Especially in an era where treatment and control arm regimens are becoming increasingly complex, adjusting the design and analysis methods to various survival curve shapes is challenging.

In silico clinical trials can provide principled predictions about possible shapes of the survival curve, including the underlying hazard rates and hazard ratios, before trial execution. We generated such

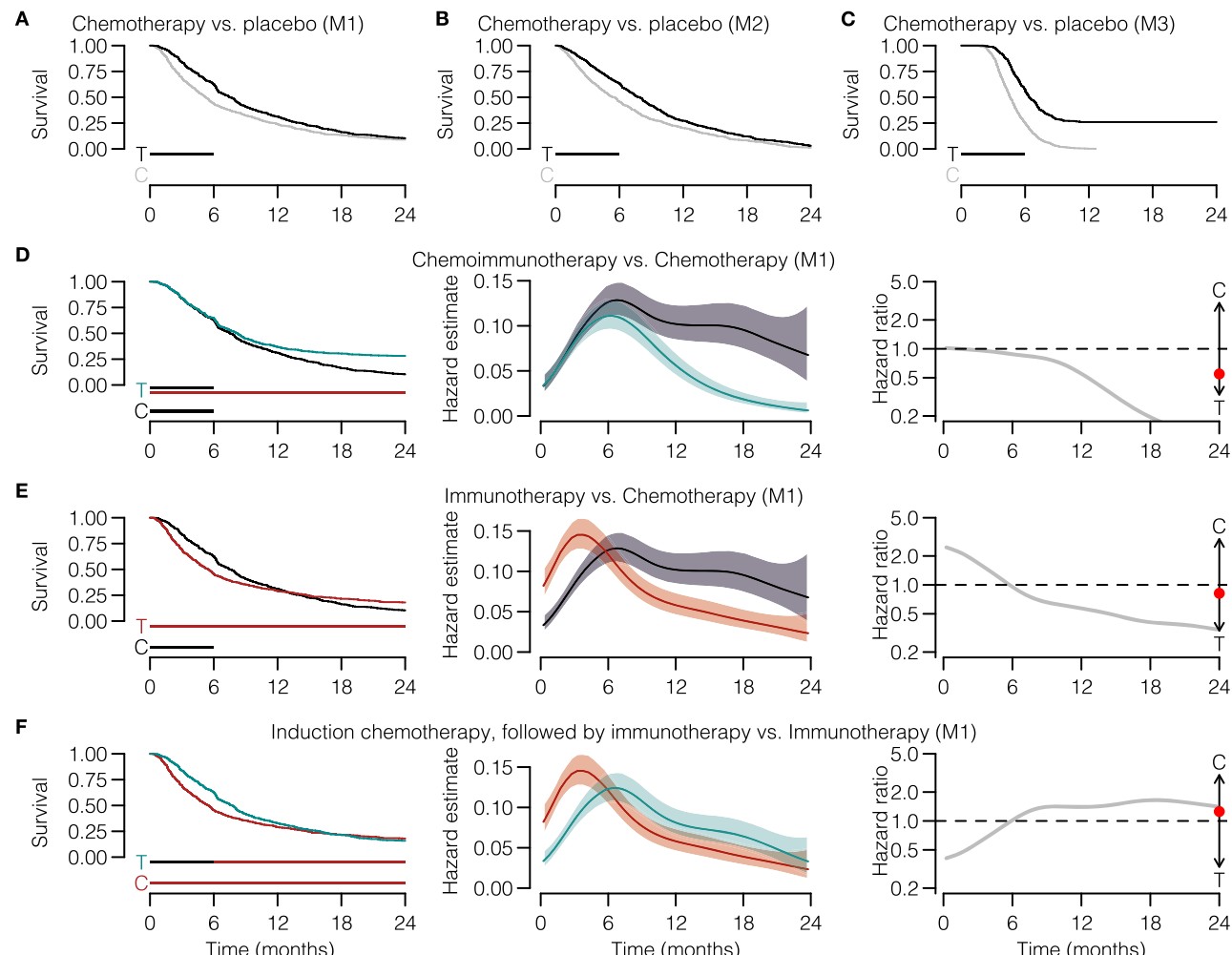

**Fig. 4 | In silico clinical trials can predict immunotherapy-specific survival patterns based on biological assumptions. A–F** Examples of 1:1 randomized trials with various (treatment) regimens (*n*=600 simulated patients per arm). **A–C** A traditional chemotherapy trial (vs. placebo) only shows a proportional hazard ratio when the biological treatment effect targets the tumor directly and remains constant over time (compare to Supplementary Fig. 2). **D** An in silico immunotherapy trial elicits typical immunotherapy-induced survival curve shapes (i.e., delayed curve separation) and violates the proportional hazard assumption. **E, F** More intricate treatment or control regimens – (**D**) immunotherapy+chemotherapy-placebo vs. chemotherapy + immunotherapy-placebo, or (**E**) induction chemotherapy followed by immunotherapy vs. immunotherapy – induce more complex survival patterns, including (**E**) crossing survival curves or (**F**) only a temporary separation of the survival curves. Horizontal bars underneath the survival curves indicate the duration of the treatment effect (T=treatment, C=control). The red dot in column three indicates the hazard ratio averaged over the entire trial. Lines and shading in middle column: estimated hazards and 95% CIs (see Methods). Source data are provided as a Source Data file.

survival predictions using the models – fitted to the CA184-024 data (Table 3) – and changed the treatment effect parameters according to the simulated scenario. A traditional scenario would be a trial in which patients are randomized 1:1 to mono-chemotherapy or placebo. Given the direct chemotherapy effect, the PHA is generally assumed to hold for these trials. An in silico trial in which chemotherapy reduces the tumor growth rate for the entire trial duration indeed replicates these assumptions (Supplementary Fig. 2): the survival curves separate from the start of the trial, and the hazard ratio remains roughly constant over time. However, what happens if the chemotherapy effect does not last for the entire trial but for – maybe more realistically–6 months? For M1 and M2, the initial proportional separation of the survival curves is followed by a parallel decay and eventual convergence of both curves, leading to an early but transient survival benefit for the chemotherapy arm (Fig. 4A, B). For M3, the chemotherapy effect estimated from the CA184-024 data is more profound and instead induces a permanent response (Fig. 4C). Hence, substantial deviations from the PHA are observed in all cases, even for seemingly simple chemotherapy trials. Also, a violated PHA becomes immediately

apparent when considering a more contemporary scenario of immunotherapy combined with chemotherapy compared to chemotherapy alone: through approximately the first six months, the hazard rates remain constant over time, but after that, they start to decline in the immunotherapy group (cyan line), yielding a non-constant hazard ratio over time (Fig. 4D).

The flexibility of in silico trials lies in their ability to incorporate complex treatment regimens. For example, let us assume one would be interested in estimating the survival curves and underlying hazard ratio over time of an immunotherapy+placebo-chemotherapy vs. chemotherapy+placebo-immunotherapy trial (Fig. 4E) or a trial with induction chemotherapy followed by immunotherapy vs. immunotherapy (Fig. 4F). Mechanism-based immunotherapy trials translate biological assumptions regarding the disease and treatment effects into survival curves (including hazard ratio estimates). The resulting survival curve shapes, such as crossing survival curves (Fig. 4E) or a temporary curve separation (Fig. 4A, B, F), may be hard to predict otherwise and can be detrimental to the trial outcome if addressed appropriately.

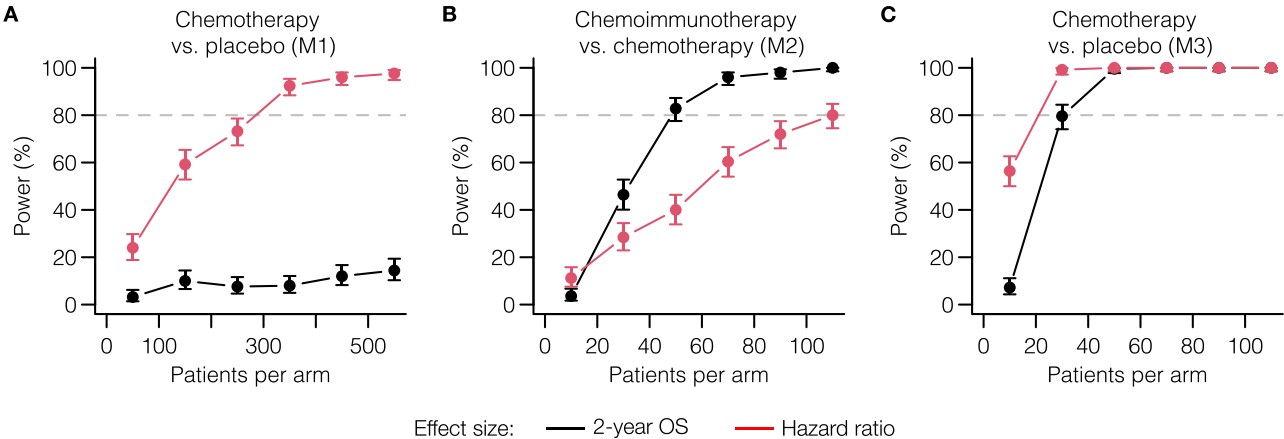

**Fig. 5 | Immunotherapy-specific survival curve shapes critically determine a trial's power to detect different treatment effects.** We analyzed the power of in silico trials to detect a difference in 2-year OS (black lines) or a hazard ratio not equal to 1 (red lines) for (**A**) chemotherapy vs. placebo (transient effect, M1); (**B**) chemoimmunotherapy vs. chemotherapy (M2); and (**C**) chemotherapy vs. placebo (long-term effect; M3). Choosing an inappropriate effect size for the response pattern at hand leads to a significant reduction in trial power, greatly reducing the probability of success. The chemotherapy effect for M3 was set to a 20% reduction in growth rate to simulate a more subtle effect; all other parameters were set to the values fitted to the CA184-024 data. Points and error bars: estimated proportions and 95% CIs from binomial tests ($n = 250$ simulated trials per point). Source data are provided as a Source Data file.

We emphasize that different models can generate different predictions depending on model assumptions and parameters, as seen in our chemotherapy vs. placebo examples (Fig. 4A, B, C). Conversely, however, even substantially different models can agree on the essential aspects of the predicted survival curves. For example, despite their differences, our three models all predict the characteristic delayed curve separation of immunotherapy trials (Fig. 4D, Supplementary Fig. 3, Supplementary Fig. 4).

## Using in silico trials to select the treatment effect metric

A key design decision in a clinical trial is which effect size metric to use to define treatment success. Two common choices are the overall hazard ratio, which is affected by the entire survival data of the trial, and a survival endpoint such as 2-year overall survival (OS), which only depends on the specifically defined time-point. When there is no solid clinical rationale to prefer one effect size measure over the other, statistical considerations such as power become important. To investigate the consequences of choosing hazard ratio or 2-year OS as the study effect size in different immunotherapy scenarios, we determined the power of in silico trials by conducting simulations at varying study population sizes.

A potential advantage of using the hazard ratio is its use of the entire survival curve, which can increase power when the PHA is met and detect transient effects even if the PHA is not met. Indeed, when investigating the power of the transient chemotherapy effect generated by model M1 (Fig. 5A), we found the power to be much greater when using the hazard ratio compared to the power to detect the minimal difference in survival still found after 2 years. The opposite was true when investigating the chemoimmunotherapy vs. immunotherapy scenario using M2 (Fig. 5B): the power of trials that used the hazard ratio lagged far behind the power to detect a 2-year survival endpoint, as a consequence of the considerable violation of the PHA in this scenario. Indeed, when considering the persistent chemotherapy effect generated by model M3 (Fig. 5C), a scenario with a substantially lower variation of the hazard ratio, we found the power to be more comparable, although the hazard ratio still had a meaningful advantage. When using M3 to investigate the chemoimmunotherapy vs. immunotherapy scenario, the choice of endpoint made hardly any difference (Supplementary Fig. 5).

These results illustrate the critical importance of choosing an appropriate effect size to measure the clinical outcome, which in turn

strongly depends on the shape of the survival curves. For established treatments, investigators can rely on their experience or published results to make an appropriate choice; however, the expected survival curve shape might be very uncertain for novel immunotherapies or combinations of existing immunotherapies. In such cases, running various in silico trials would help investigators prepare for different plausible scenarios and choose a robust trial design. In our examples, the models agreed that hazard ratio would be a suitable effect size for a chemotherapy vs. placebo trial even if the PHA does not entirely hold, whereas 2-year OS would be appropriate for the chemoimmunotherapy vs. immunotherapy case (Supplementary Fig. 5).

## In silico trials can help to choose endpoints and randomization ratios

Clearly, the success rate of novel immunotherapy trials depends on more than its sample size alone. To establish an OS benefit of the treatment arm, it is crucial to analyze the trial once the data have reached a certain maturity – i.e., the treatment needs to be granted sufficient time to induce a survival benefit. We assumed that a delayed curve separation in immunotherapy trials would prolong the follow-up needed to establish an OS benefit of immunotherapy and thereby defer reaching maturity of the trial data. If the therapy is effective, data maturity can be regarded as the time point when a treatment effect can be observed. Hence, an optimal trial endpoint would be the earliest time at which this treatment effect can be detected with sufficient power. Therefore, we analyzed the power of differently-sized trials with respect to their OS endpoint. Herein, we distinguished trials that were subject or were not subject to a delayed curve separation (immunotherapy and chemotherapy, respectively). In a classic chemotherapy trial, the treatment effect translates directly to a survival benefit in the treatment arm – the survival curves separate from the start. Therefore, the highest power is obtained after the total duration of the treatment effect (Fig. 6A, panel 1). In this case, the treatment effect lasts for six months, leading to the 6-months OS as the endpoint with the highest power. The delayed curve separation in immunotherapy trials renders it futile to analyze OS data early in the trial (Fig. 6B, panel 1). A practical ramification is that in the presence of a delayed curve separation, the trial requires a sufficiently long follow-up and an adequate size to gain power and detect immunotherapy-specific treatment effects. Mechanism- and simulation-based power calculations with in silico trials can consider these specific

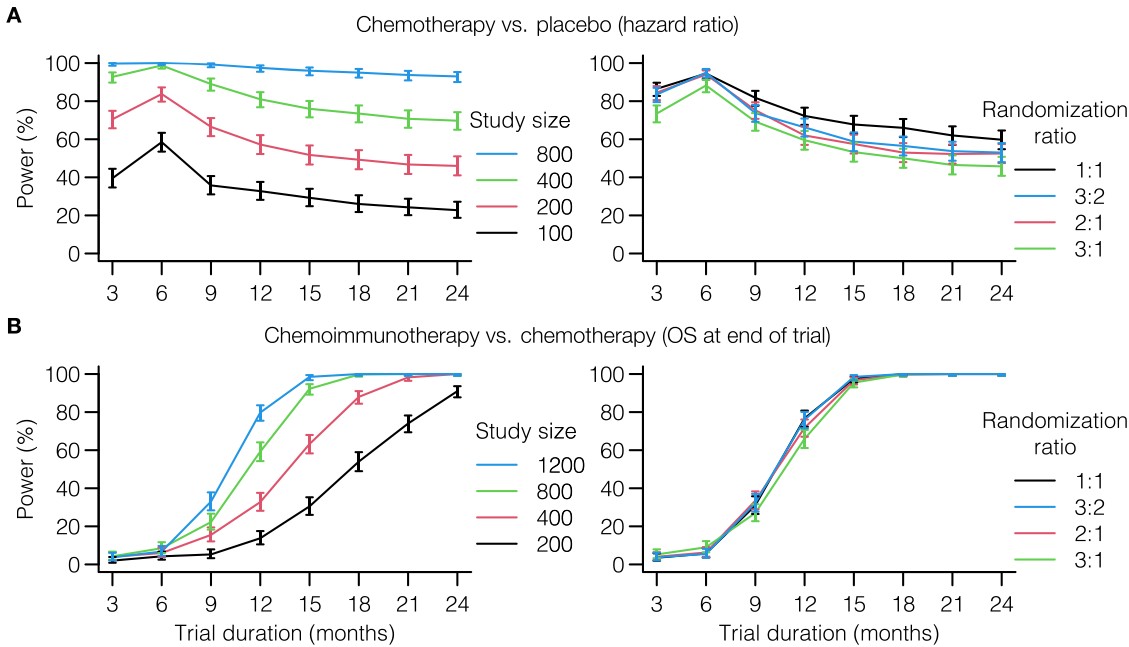

**Fig. 6 | In silico trials guide decisions on OS endpoints and randomization ratios of upcoming immunotherapy trials. A, B** In silico trials can be used to find the optimal endpoint (panel 1) or randomization ratio (panel 2) of novel trials. **A** Since the survival curves in classical chemotherapy trials separate from the trial onset, the highest power – and optimal endpoint – is obtained at the end of the treatment interval (i.e., after six months in this example; see Fig. 4A). Although less influential, a similar observation can be made for randomization ratios (study size panel 2: 300 patients). **B** Delayed curve separation in immunotherapy trials emphasizes that a premature final analysis of the primary OS endpoint is detrimental to the trial outcome. These trials permit validating the pre-specified survival outcomes of novel trials a priori. Commonly selected randomization ratios do not seem to be heavily influenced by immunotherapy-specific response patterns (study size panel 2: 1200 patient). Trial characteristics are similar to Fig. 4A, D. All simulations performed using M1. Points and error bars: estimated proportions and 95% CIs from binomial tests ($n = 400$ simulated trials per point). Source data are provided as a Source Data file.

survival curve features when determining the sample size for upcoming trials.

Given the observation that both the size of an immunotherapy trial and its endpoint heavily influence the probability of finding the survival benefit of interest, we presumed that increasing the size of the treatment arm – i.e., an unequal randomization scheme – would similarly affect the power. Instead of varying the study size, we now varied the randomization ratio. Interestingly, while the power logically depended on the OS endpoint, the randomization ratio did not greatly affect the power (second panel of Fig. 6A, B). Considering that an unequal treatment allocation may provide ethical benefits, we confirm that the randomization ratio in immunotherapy trials is of secondary importance compared to study size or primary endpoint.

In summary, our in silico immunotherapy trials replicate existing insights from trial design as to how violation of the PHA affects power and analysis choices. Our ability to directly translate biological assumptions on treatment mechanisms into survival curve shapes allows the trialist to reason deliberately about whether such violations of the PHA would or would not be expected in their specific trial design and how the problem could be addressed if it arises.

**Simulating the effects of interim analyses**

We have observed a clear trade-off between the power of an immunotherapy trial on the one hand, and the primary OS endpoint, and correspondingly the data maturity, on the other. Luckily, the two are not entirely mutually exclusive: interim analyses have been developed for ethical purposes to establish positive or harmful treatment effects early. However, there is a catch: the necessity to control for multiple testing at each interim analysis lowers the significance threshold on the final analysis to maintain the same overall type I error rate. This raises the question: "How many interim analyses should you plan, and when should you plan them?" Again, well-founded answers to such questions

can be obtained with the help of in silico immunotherapy trials. To illustrate this, we used M1 to simulate 1000 immunotherapy trials with 1200 patients per trial, randomized 1:1 over immunotherapy with a strong treatment effect or a placebo (Fig. 7A). In the absence of interim analyses, the vast majority of the trials are predicted to end up positive. Adding interim analyses (O'Brien-Fleming approach) to the equation induces a trade-off. On the one hand, increasing the number of equally-spaced interim analyses increases the probability of early detecting a positive treatment effect (e.g., approximately 40% of the trials are positive after 18 months in the case of three interim analyses; Fig. 7A). On the other hand, the overall probability of ending up with a negative trial due to more stringent analyses (i.e., less power) also increases slightly, especially in the case of immunotherapies with a weaker treatment effect ($\pm$ 88% without an interim analysis vs. $\pm$ 86% with three interim analyses; Fig. 7B). In an actual trial, the latter needs to be corrected by including additional patients to maintain the pre-planned power. Furthermore, we observe that the timing of the interim analysis is crucial. Whereas an interim analysis at 18 months provides additional value to the trial, interim analyses before 16 months are predicted to be wasteful due to non-proportional hazards and less mature data. As a control, we simulated trials without any treatment effect. By design, approximately 95% of the trials should end up negative irrespective of the number of interim analyses, which indeed seemed to be the case (Fig. 7C). Logically, the weaker the treatment effect, the higher the probability of erroneously finding a harmful treatment effect – a characteristic that the simulation also exhibits (Fig. 7B, C).

## Discussion

Over the past decade, tumor-immune dynamics have been investigated extensively with in silico models. In the early days of cancer immunotherapy, these modeling efforts focused – next to chemotherapy[37] – on cellular immunotherapy[38,39]. More recently, the

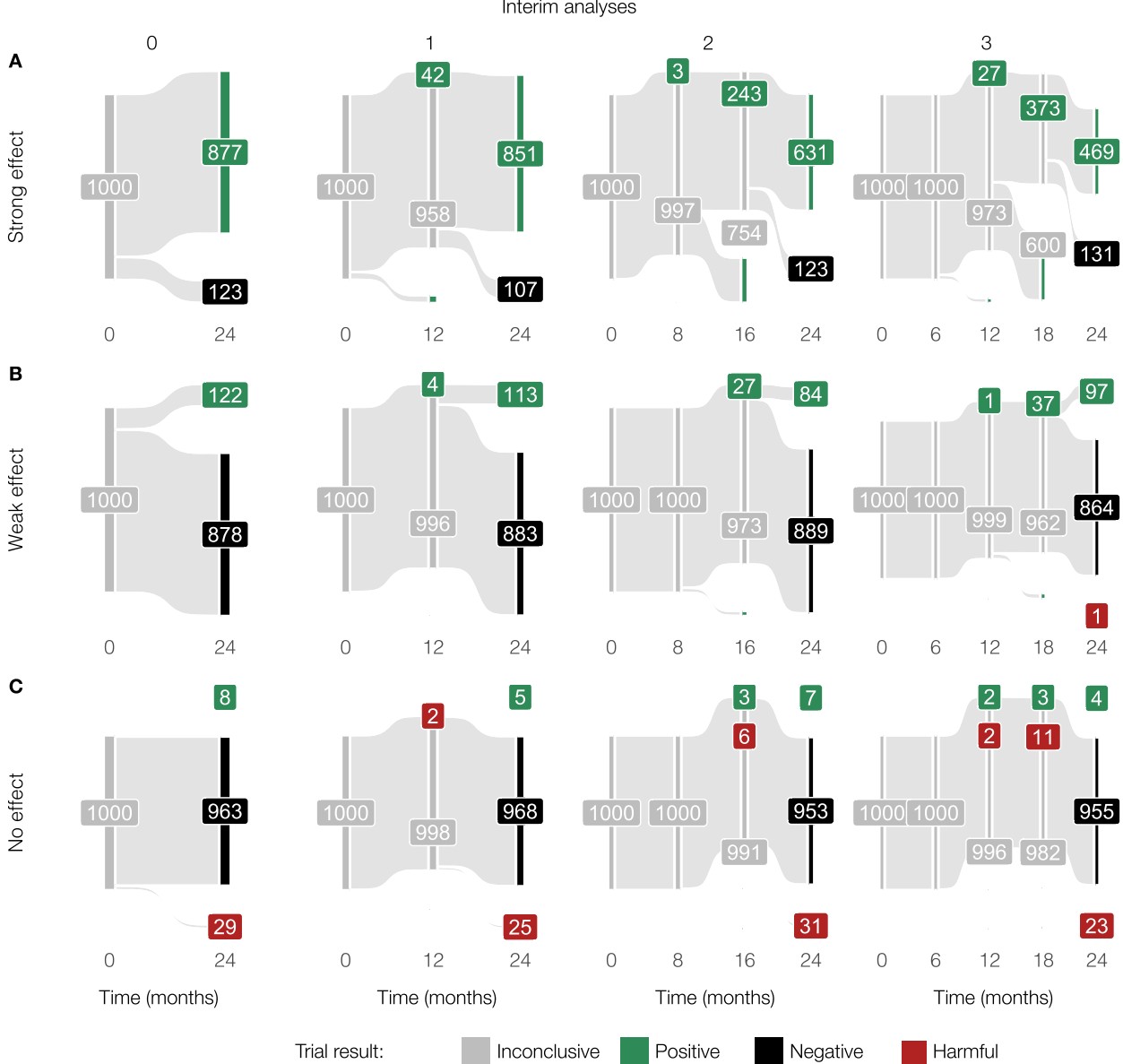

**Fig. 7 | A priori scrutiny of the interim analysis plan to evaluate possible advantages and disadvantages of timed additional analyses during the trial. A** In the case of immunotherapy with a potent effect, in silico trials help develop a rationale for the timing of the interim analyses. In these simulations, while an interim analysis at 12 months might not add value to the trial, analyses after 16 and 18 months, respectively, have a probability of approximately 25% and 40% to lead to early stopping with a positive result. **B** Multiple interim analyses can reduce the probability of confirming the desired treatment effect in case of a weak immunotherapy effect. **C** In the absence of any treatment effect (a control scenario), the number of interim analyses does not heavily influence the trial outcome. Each trial simulation contains 1200 patients (randomization ratio 1:1) to ensure adequate power of the trial. Trials are analyzed with a proportions test (Pearson's chi-squared test). Treatment effect (fold increase of the T cell killing rate): strong=12, weak=4, no effect=0 (see Methods). All simulations were performed using M1. Source data are provided as a Source Data file.

field has addressed ICI therapy (e.g.,[33,40,41]). Models of tumor-immune dynamics have been applied to study pharmacokinetic and therapy dynamics (PK/PD), treatment effects (including mechanism(s) of action, optimizing dosing regimens), treatment combinations, toxicity, biomarker prediction, drug resistance, and drug discovery (see reviews on these topics[38,42–47]). These extensive modeling efforts by the Mathematical Oncology community have created a rich and valuable methodological resource. The goal of the work described in this paper is to tap into this resource for the purpose of clinical trial design. Once parameterized, a mathematical model can predict likely outcomes of treatments for individual patients; using such a model for trial design requires considering heterogeneity between patients and translating these into likely survival curve shapes for each arm of the trial.

In this study, we leveraged mathematical models to perform cancer immunotherapy trials in silico, predicting survival and response profiles of various treatment regimens. Complementary to conventional design methods, in silico trials provide the ability to investigate the implications of a researcher's biological (as opposed to statistical) hypotheses about a drug's mechanism of action for the design, conduct, analysis, and outcome of clinical trials. When comparing the simulated outcomes to actual immunotherapy trial outcomes, we showed that in silico trials are suited to translate complex biological mechanisms (such as those observed during the treatment of patients with ICI) into realistic trial outcomes. Crucially, regardless of the model, the survival curves from these mechanism-based simulations reflected two pivotal components often found in immunotherapy

trials: a delayed curve separation and a plateauing tail of the survival curve at later stages of the trial. In line with genuine immunotherapy trials, we find that these immunotherapy-specific response patterns differ considerably from chemotherapy. Our findings confirm that diversity in survival curves profoundly impacts the outcomes of immunotherapy trials[48]. Consequently, these features need to be considered when deciding on the sample size, endpoint, randomization ratio, and the number and timing of interim analyses of a novel cancer immunotherapy trial.

In silico clinical trials are gaining popularity in medicine. Such trials enable investigating, among others, how novel drugs, treatment schedules, dosing regimens, and inter-patient heterogeneity affect the outcome of a clinical trial[49]. In silico clinical studies have a wide range of applicability from pediatric infectious[50] and orphan diseases[51] to diabetes[52], inflammatory autoimmune diseases[53], traumatic injury[54], psychiatric illness[55], and cancer. In oncology, several in silico clinical trials involving chemotherapy and tyrosine kinase inhibitors have been performed[56,57]. Moreover, with the onset of checkpoint inhibitors, in silico immunotherapy trials have gained interest, leading to in silico trials with anti-CTLA-4-antibodies and anti-PD-(L)1 antibodies[58–60]. In contrast to these earlier studies, our work does not primarily focus on the treatment itself but on the design of clinical trials intended to show efficacy of a given treatment. In that sense, our work is more comparable to statistical simulation studies aiming to calculate the sample size and power of clinical trials[61–63]. However, purely statistical approaches lack a direct link to the underlying biological disease mechanism. As we have seen, such a link is critical in the case of immunotherapy trials, which tend to violate common statistical assumptions of trial designs such as PHA. Therefore, an interdisciplinary approach to trial design that combines these two perspectives – modelling and statistics – could be especially beneficial in the oncoimmunology field.

In silico clinical trials are applicable in several settings. First, they provide the means to verify clinical trial and treatment assumptions before investing extensive amounts of work and funds into the development and execution of a clinical trial and can, thereby, function as a proof of principle of the soundness of the hypotheses for an upcoming trial. Scrutinizing each aspect of the trial design might lead to better design decisions and reduce unanticipated outcomes. Moreover, this mechanism-based approach does not necessitate a deep understanding of complex mathematical theorems; instead, it requires a biological understanding of a disease. This mechanistic basis is intuitive, which benefits the communication between clinical doctors and biomedical researchers on the one hand and statisticians and clinical trialists on the other. Additionally, in silico trials might serve as excellent educational tools. The ability to simulate a wide range – from basic to highly advanced–research questions can be exploited in teaching activities for entry-level clinicians to experienced trialists. A final implication, which holds for any trial simulation, is that they may provide some degree of insight when conventional clinical trials are unfeasible due to practical or ethical constraints (e.g., clinical trials in rare diseases, pediatrics, or critical care medicine).

Nonetheless, in silico clinical trials have to be considered in light of some limitations. The most critical limitation is universal to any scientific model, whether in vitro, in vivo, or computational: the immunotherapy trial outcomes depend heavily (if not entirely) on the biological assumptions of the model, meaning that incorrect interactions or erroneous parametrization of the model can lead to inaccurate predictions. The parameterization, in particular, might pose a problem: given the often novel treatment mechanisms, data to fine-tune the parameters of the model accurately might be scarce. In these cases, the simulation itself can be used as a sensitivity analysis to assess to what extent a certain parameter range, or the structure of the model itself, influences the robustness of the predictions. Our use of three different models in this paper can be seen as such a type of sensitivity analysis; indeed, despite the major differences, it was reassuring to observe that the models often agreed when it came to the critical qualitative aspects of the predicted survival curves.

In addition, while ODE models can be rather simple and intuitive to understand, translating biological principles into an ODE model and implementing it into a simulation requires thorough knowledge of computational methods, potentially limiting its widespread applicability. To address these limitations, we have made our model implementations available as (1) an interactive website that can be used without installing any software and without any programming knowledge (https://computational-immunology.org/models/immunotherapy-trials/); (2) an R package allowing to run simulations without requiring knowledge of ODEs and their solutions.

In summary, in silico cancer immunotherapy trials offer a versatile approach to simulate immunotherapy trials based on biological assumptions. As a simulation tool, they facilitate the scrutiny of trial design decisions to optimize the probability of a successful immunotherapy trial and contribute to high-quality research for cancer patients.

## Methods

### Ethics
No human subjects participated in this study, and we did not analyze any identifiable data. Survival data from three published, completed clinical studies was retrospectively analyzed: the NCCTG cohort[34], the CA184-024 trial[35] (NCT00324155), and the CheckMate 066 trial[36] (NCT01721772). We had no access to individual patient data from these cohorts and only analyzed the survival times. For the NCCTG cohort, these data are available publicly as part of the R package "survival"[64]; for the CA184-024 and CheckMate 066 trials, we estimated the survival data by digitizing published figures as described below.

### Mechanism-based models of the tumor microenvironment
We implemented three ODE models of tumor-immune interactions: one from our previous work[27] and two by other authors[28,29]. We first describe the common aspects of the models, then explain the differences and show the model equations. All models describe cancer onset and progression, and we initialize each model by seeding a single growing tumor cell. This tumor cell divides, leading to a proliferating mass of tumor cells. The parameter $\rho$ controls the grows rate. Within the tumor microenvironment, an anti-tumor immune response induces cytotoxic T cells to kill tumor cells at rate $\xi$. Intratumoral T cells die at a rate $\delta$. In these models, the rate at which T cells are activated and/or proliferate depends initially on the tumor size: an early-stage microscopic tumor presents fewer antigens than a larger – but still small – tumor. However, antigen presentation saturates as the tumor grows further (scaling factor $T/(g + T)$ in Equations (3),(4),(3),(9)). Thus, four model parameters are shared between the models. Depending on the parameter values, it is possible that the immune response eliminates the tumor or that the tumor escapes and grows in an uncontrolled fashion.

We now discuss the model equations and parameters. In all models, we denote the number of tumor cells by $T$ (Eq. (1),(5),(8)) and the number of intratumoral T cells by $I$ (Eq. (2),(6),(9)). Compared to their original versions, variables and parameters in the equations below have been renamed, and the units of some parameters scaled to make the models easily comparable.

Model M1 is based on our previous work[27] and has the following equations:

$$\frac{dT}{dt} = \rho T^{\frac{3}{4}} - \xi I \frac{T}{1 + \frac{I}{h_I} + \frac{T}{h_T}} \tag{1}$$

$$\frac{dI}{dt} = m_S S - \delta I \tag{2}$$

$$\frac{dS}{dt} = \left(\frac{T}{g+T}\right)(\alpha_N N + p_S S) - m_S S \tag{3}$$

$$\frac{dN}{dt} = -\left(\frac{T}{g+T}\right)\alpha_N N \tag{4}$$

We implemented a tumor growth rate that is slower than exponential growth. This is a common modeling choice based on data and the biological premise that a growing tumor needs to sustain itself with nutrients. A common method to implement a sub-exponential growth, which we adopt here, is to raise the number of tumor cells to the 3/4th power to obtain the number of actively dividing cells[65]. We had previously modeled slightly faster-growing tumors using the less common power 4/5th[27]. However, given that the other two models already implement faster-growing tumors, we here use the more common, slower one. The killing of tumor cells is implemented using a double saturation model[66] parameterized as proposed by Gadhamsetty et al.[67] (Michaelis constants $h_T$ and $h_I$). The double saturation model reflects that T-cell killing of tumor cells takes hours[68]. The immune cells within the tumor microenvironment originate from tumor-draining lymph nodes, where naive cytotoxic T cells ($N$, Equation (4)) turn into activated T cells ($S$, activation rate $\alpha_N$; Eq. (4)). Activated T cells proliferate at rate $p_S$ and migrate to the tumor microenvironment to become infiltrating T cells ($I$). The migration step leads to a slight delay between T cell activation and tumor cell killing on the order of days ($m_S = 1\,\text{day}^{-1}$). If desired, the distinction between lymph node and tumor microenvironment sites could be removed for simplicity, given that the migration takes place on a faster timescale than the immune response.

Model M2, proposed by Tsur et al.[28], conceptually differs from M1 in five aspects. First, its tumor growth is unrestricted exponential. Second, the anti-tumor response saturates with increasing numbers of tumor cells but not with increasing numbers of T cells. Third, it explicitly represents antigen-presenting cells, called $A$ (Equation (7)), which are recruited at rate $\alpha_A$ in response to the tumor growth. Fourth, its T cells do not proliferate but are produced at a capped rate. Fifth, it does not distinguish between T cells in the lymph node and intratumoral T cells; as mentioned above, this is likely not critical. The model equations are as follows:

$$\frac{dT}{dt} = \rho T - \xi I \frac{T}{1 + \frac{T}{h_T}} \tag{5}$$

$$\frac{dI}{dt} = \alpha_e A - \delta I \tag{6}$$

$$\frac{dA}{dt} = \alpha_A \frac{T}{g+T} - \delta_A A \tag{7}$$

Model M3 was recently proposed by Bekker et al.[29]. It has two equations representing tumor cells and T cells. It resembles M2 in that tumor growth is initially exponential, but there is a maximum capacity for tumor cells (logistic growth). Killing dynamics follow a "mass-action law" (i.e., there is no saturation of the killing rate like in M1 and

M2). Further, it includes a term for tumor size-dependent T-cell exhaustion. This modeling choice leads to oscillating numbers of T cells and tumor cells in many parameter regimes. The model equations are as follows:

$$\frac{dT}{dt} = \rho T(1 - T/\beta) - \xi I T \tag{8}$$

$$\frac{dI}{dt} = \alpha_A + p_I I \frac{T}{g+T} - \delta I - \epsilon I T \tag{9}$$

We emphasize that M3 has been presented by Bekker et al.[29] as an abstraction of the general mechanisms underlying immunotherapy similar to M1; neither model claims to fit specific time-resolved data. Nevertheless, we included it as we were interested in the impact of the different modeling choices.

## Model parameters

Table 1 shows an overview of the parameters in the three models. Four parameters appear in every model, but note that this does not necessarily mean that the parameters can be interpreted in the same way. For example, in a model where killing saturates in a scenario where there are many more tumor cells than T cells (M1 and M2), the same value of the killing rate will lead to less effective killing than in a model where there is no such saturation (M3). Other parameters are model-specific. To improve the inter-model comparability and reduce the potential for over-fitting, we left the parameters in all models fixed except the tumor growth rate $\rho$, which we varied to obtain heterogeneous patient populations.

Parameter values for M1 and M2 were taken from earlier publications[27,28], where the biological reasoning underlying these values is explained, and references are provided. Differences in model structure, and in experimental data being referred to, yield extensive variation in parameter values (Table 2). The variation in $\rho$ is just a consequence of the different tumor growth models, which give a different meaning to the parameter in each model. Despite the

## Table 1 | Overview of parameters used in the three models

| Model(s) | Symbol | Meaning | Unit |
|---|---|---|---|
| M1,M2,M3 | $\rho$ | Tumor proliferation rate | day⁻¹ |
| M1,M2,M3 | $\delta$ | T cell death rate | day⁻¹ |
| M1,M2,M3 | $\xi$ | T cell killing rate | day⁻¹ cell⁻¹ |
| M1,M2,M3 | $g$ | Amount of tumor cells at which antigen presentation is half-maximal | cell |
| M1,M2 | $h_T$ | Michaelis constant for tumor-dependent killing saturation | cell |
| M2,M3 | $\alpha_A$ | T cell or antigen presenting cell influx | cell day⁻¹ |
| M1 | $h_I$ | Michaelis constant for T cell-dependent killing saturation | cell |
| M1 | $m_s$ | T cell migration rate | day⁻¹ |
| M1 | $p_s$ | Proliferation rate of T cells in lymph nodes | day⁻¹ |
| M1 | $\alpha_N$ | Activation rate of naïve T cells | day⁻¹ |
| M2 | $\alpha_e$ | Production rate of intratumoral T cells | day⁻¹ |
| M2 | $\delta_A$ | Death rate of antigen-presenting cells | day⁻¹ |
| M3 | $p_I$ | Proliferation rate of intratumoral T cells | day⁻¹ |
| M3 | $\beta$ | Maximum number of tumor cells the body can sustain | cell |
| M3 | $\epsilon$ | Rate at which tumor cells exhaust T cells | day⁻¹ cell⁻¹ |

Four parameters are shared between all models. The number of parameters is the largest for M1 at 9 parameters, followed by M2 and M3 (8).

**Table 2 | Fixed parameter values used in our simulations**

| Parameter | $\rho$ | $\delta$ | $\xi$ | $g$ | $h_T$ | $\alpha_A$ | $h_I$ | $m_S$ | $p_S$ | $\alpha_N$ | $\alpha_e$ | $\delta_A$ | $p_I$ | $\beta$ | $\epsilon$ |
|---|---|---|---|---|---|---|---|---|---|---|---|---|---|---|---|
| M1 | 5 | 0.019 | 0.001 | 10000000 | 571 | | 571 | 1 | 1 | 0.0025 | | | | | |
| M2 | 0.045 | 0.178 | 0.00000134 | 92330 | 60095000 | 2073.5 | | | | | 0.8318 | 0.231 | | | |
| M3 | 0.045 | 0.019 | 0.0000000001 | 10000000 | | 2073.5 | | | | | | | 0.05 | $1.1 \cdot 10^{12}$ | $10^{-12}$ |

All values except $\rho$ are taken from previous work, and are kept constant in all simulations. The value of $\rho$ is allowed to vary between simulated patients to account for heterogeneity, and the distribution of $\rho$ is fitted to real data. The values of $\rho$ shown here were those used to generate Fig. 2.

differences, the values actually lead to comparable growth kinetics. The biggest quantitative differences are in the killing kinetics. M2's killing rate $\xi$ is three orders of magnitude smaller than M1's, but M2 compensates for this by saturating the killing at a number of tumor cells that is five orders of magnitude higher than M1's. Overall, the number of cells being killed when the immune system is active and the tumor exceeds the diagnosis threshold is comparable across the models.

M3 was not explicitly parameterized by the authors[29]. Therefore, we set its parameters to the same values as in M1 or M2 as much as possible. For instance, because both M2 and M3 contain essentially unrestricted exponential growth of the tumor cells until M3 approaches the carrying capacity, we used the value for the tumor growth rate in M2 for M3. For the killing rate, we used a value that gave similar killing speed as M1 for tumors containing $10^9 - 10^{10}$ T cells. Note that due to the saturation term in M1, the killing is faster in M3 for larger tumors and slower for smaller tumors. Two parameters, the T cell exhaustion rate and the carrying capacity, were unique to M3. We set both to values that lead to a small influence of the corresponding terms on the simulation result and obtained comparable kinetics to the other two models at those parameter settings.

### Simulating untreated disease and treatments in individual patients

Using ODE models, we can implement different cancer immunotherapies in two general ways: (1) by changing model parameters; (2) by adding or removing cells at a certain time.

Using this ODE model, we simulated cancer development and disease trajectories in patients. We extensively varied the tumor properties (i.e., the tumor growth rate, the growth rate decline, and the decline decay rate) between patients to generate interpatient variation in disease courses.

Each patient is simulated from cancer onset (i.e., malignant transformation of the first cell) for up to ten years. As argued previously[27], we start from a diagnosis threshold of a tumor mass of $65 \times 10^8$ cells, corresponding to the size at which common malignancies are diagnosed[69–71]. The lethal tumor burden is set to $10^{12}$ tumor cells (a tumor volume of approximately 10.6 dm³). Since we expect both thresholds to vary considerably between patients, depending, for example, on the timing of doctor visits or a tumor's location, we implement them as random variables that change with every simulation. Specifically, every threshold is drawn from a log-normal distribution with a $4\sigma$ range of one order of magnitude. The upper $2\sigma$ point (95.45% quantile) is set to $65 \times 10^8$ for diagnosis and $10^{12}$ for death.

Disease trajectories of patients with cancer can be steered with therapy. In our model, treatment is implemented by changing the model parameters once the tumor exceeds the diagnosis threshold, as we assume this is when treatment starts. Given their prominent roles in many oncological treatment plans, we included immune checkpoint inhibitors (ICI) and chemotherapy in the models. Both treatments function through their primary modes of action. ICI are implemented by increasing the killing rate of cytotoxic T cells (i.e., the parameter $\xi$ in M1 and M3. In M2, it is implemented by increasing the T cell activation rate $\alpha_A$ and decreasing the death rate $\delta$; for simplicity, we

restrict this such that the fold increase of $\alpha_A$ equals the fold decrease of $\delta$. These changes are implemented directly after diagnosis and remain active for the rest of the simulation unless stated otherwise. The duration and potency of the ICI treatment (as measured by the magnitude of the change of the affected parameters) eventually determine patient outcome.

In patients treated with chemotherapy, the immune system is still present; however, it is not boosted (as is the case during ICI treatment). Hence, the T cells are not potent enough to curb tumor growth. We implement the cytotoxic capacity of chemotherapy in the models uniformly by reducing the tumor growth rate (parameter $\rho$) to a smaller number. Again, the duration and potency (as measured by the reduction in tumor growth rate) determine patient outcome. By default, the treatment duration for ICI and chemotherapy are two years and six months, respectively.

### Simulations of patient cohorts and parameter fitting

To generate heterogeneous patient populations, we draw each patient's growth rate parameter $\rho$ from a log-normal distribution. Depending on the parameter, the simulated patient's tumor may clear spontaneously; such results are discarded (rejection sampling). When the tumor reaches the diagnosis threshold, we apply ICI, chemotherapy, a combination of chemotherapy and ICI, or we leave the patient untreated (i.e., a placebo treatment). Therefore, each patient cohort (Fig. 3) is characterized by two to four parameters: mean and standard deviation of the log tumor growth rate, immunotherapy treatment effect size, and chemotherapy treatment effect size. These two to four parameters can be fitted to a given dataset.

Due to the stochastic nature of our model, we used an approximate Bayesian computation / sequential Monte Carlo (ABC-SMC) algorithm[72] to fit the parameters. As the test statistic for ABC-SMC, we used the root mean squared difference (RMSD) between model-predicted and data-estimated survival curves (i.e., Kaplan–Meier curves) evaluated for each month in a 2-year time window upon diagnosis. We set the sample size for generating the model-predicted survival curve to the same number of patients that is contained in the data being fitted. Figure 3 and Supplementary Fig. 1 show, for each model, the simulation that achieved the lowest RMSD to the target data during each ABC run.

We applied the ABC-SMC algorithm to all three patient cohorts shown in Fig. 3. When examining the posterior distributions of the parameters, we found that a wide range of chemotherapy effect values achieved comparable RMSD values for each model – which is not surprising, given that a higher baseline growth rate combined with a higher chemotherapy effect leads to similar predicted tumor growth as a lower baseline growth rate combined with a lower chemotherapy effect. We, therefore, performed a further set of fits to the CA184-024 data where we kept the chemotherapy effect values fixed at 0.6 for M1 and M2 and at 0.75 for M3, respectively – values that were chosen to obtain comparable and realistic impacts of chemotherapy on the 2-year OS curves (Fig. 5A), and were plausible given the posterior distributions. We then again fitted the remaining three parameters to the CA184-024 data using ABC-SMC. By estimating the mode of the joint posterior distribution using kernel density smoothing, we obtained the parameter values shown in Table 3.

**Table 3 | Fitted parameter values used in Fig. 5–7**

| Parameter | $\rho$ (log$_{10}$ mean) | $\rho$ (log$_{10}$ sd) | chemotherapy effect | immunotherapy effect |
|---|---|---|---|---|
| M1 | 2.54 | 1.01 | 0.6 | 12.15 |
| M2 | −3.94 | 1.29 | 0.6 | 218.75 |
| M3 | −3.49 | 0.37 | 0.75 | 2.25 |

The chemotherapy effect parameters were fixed and the other three parameters were then fitted using Approximate Bayesian Computation as described in the methods.

## Simulating late-stage immunotherapy trials

Late-stage (i.e., phase III) clinical trials traditionally contain two arms: a control arm and a treatment arm. The control arm can be a placebo (i.e., untreated) or a standard-of-care therapy. To construct phase III in silico immunotherapy trials, we extended the simulations with treatment cohorts (mono-chemotherapy, mono-immunotherapy, chemoimmunotherapy, or induction chemotherapy followed by immunotherapy). These cohorts facilitate the comparison between various treatment regimens. The treatment cohort uses the same baseline distribution of tumor growth parameters as the control cohort. Upon reaching the diagnosis threshold, up to two different treatments are applied in each arm; patients can be treated with chemotherapy, ICI, combination therapy, or left untreated (as described above). Unless otherwise specified, the baseline distribution of tumor growth parameters was derived from the most mature, digitized data from the CA184-024 trial, as shown below[35].

The primary endpoint of the trials is the 2-year OS. Given the absence of accrual times in silico trials, the trial duration equals two years, providing each virtual patient with 24 months of follow-up at the time of analysis. If the OS endpoint is not reached for a patient (i.e., the patient's tumor burden does not reach the lethal volume within the time frame of the simulated trial), the patient is considered censored for the endpoint and regarded as such in subsequent analyses.

## Power and interim analysis simulations

To illustrate how the analysis method can affect the outcome of immunotherapy trials, we use several simulation approaches to calculate the power of trials. Power simulations were performed as follows: a varying number of clinical trials were simulated per data point. The survival data from each trial was analyzed with a log-rank test (dependent on the proportional hazard assumption) or proportions test (Pearson's chi-squared test; independent of the proportional hazard assumption), and we counted the number of positive trials (defined as $p < 0.05$). The percentage of positive trials indicates the power of the trial. A harmful trial is defined as a positive trial with an effect size that favors untreated patients.

## Data digitization & reconstruction

For some survival curves, the raw data was not available. Therefore, we extracted data points from the Kaplan–Meier curves with WebPlotDigitizer 4.6 (https://apps.automeris.io/wpd/), and individual patient data was reconstructed with the IPDfromKM package in R.

## Analyses

Analyses and visualizations were performed in R. The complete list of R packages used throughout this manuscript is provided in Supplementary Table 1. The R code used to perform all analyses is available at https://github.com/jtextor/insilico-trials. For hazard estimates in Fig. 4, Supplementary Fig. 2–4, B spline estimation as implemented in the R package "bshazard" version 1.1 was used[73].

## Reporting summary

Further information on research design is available in the Nature Portfolio Reporting Summary linked to this article.

## Data availability

All simulated data used to generate the figures is available at this paper's GitHub repository at https://github.com/jtextor/insilico-trials[74]. The digitized survival curves from the CA184-024 trial[35] and the CheckMate 066 trial[36] are also available at the same repository. The survival data of the NCCTG lung cancer cohort[34] are available publicly as part of the R package "survival" version 3.3-1[64] (and most other versions of this package). Source data are provided with this paper.

## Code availability

C++ code that implements models M1, M2 and M3, and an R package that wraps the C++ code using Rcpp[75] is available at this paper's GitHub repository at https://github.com/jtextor/insilico-trials/models/TumorImmuneModels/[74]. The R code used to perform all analyses and generate all plots shown in this paper is also available at the same repository[74]. An interactive, web-based implementation of our models, written in JavaScript and HTML, is available at the same repository and directly accessible at https://computational-immunology.org/models/immunotherapy-trials/.

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

## Acknowledgements

J.H.A.C. was funded by the Radboudumc. C.G.F. received an ERC Adv Grant ARTimmune (834618) and an NWO Spinoza grant. I.J.M.dV received an NWO-Vici grant (918.14.655). JT and A.A. were supported by a Young Investigator Grant (10620) from the Dutch Cancer Society. J.T. and G.S. were also supported by NWO grant VI.Vidi.192.084. We thank Shabaz Sultan for his help with implementing the web-based frontend for performing simulations.

## Author contributions

J.H.A.C. and J.T. conceived this study. J.H.A.C., A.A and J.T. performed the simulations and data analysis. G.S. contributed to development of the C++ code for the simulation models. J.H.A.C. and J.T. wrote the manuscript. C.G.F. and I.J.M.dV. acquired funding. C.G.F., I.J.M.dV., and N.M. contributed to supervision of the work by J.H.A.C. K.C.B.R. provided critical expertise and input on clinical trial design. All authors critically revised the manuscript for important intellectual content.

## Competing interests

The authors declare no competing interests.
