## [Peer Review File · Nature Communications]

In silico cancer immunotherapy trials uncover the consequences of therapy-specific response patterns for clinical trial design and outcomeREVIEWER COMMENTS

Reviewer #1 (Remarks to the Author):

This paper deals with a very interesting topic: the potential use of in-silico studies to guide clinical trial design of immunotherapy treatments. I liked a lot the style of the paper and fully agree with the insightful claims made by the authors in the introduction. I also agree with their conclusion that in silico trials may provide a fast and cheap approach to verify the robustness of biological assumptions underlying immunotherapy trials and help to scrutinize its design.

That said I do not think Nature Communications is the right place for this paper. First of all other versions of the model (and a similar methodology) has been used in previous works by the same authors. Also, in-silico studies have been used recently in the literature to simulate the outcome (and suggest improvements) of treatments with either chemotherapy, tyrosine kinase inhibitors and anti-CTLA-4-antibodies and anti-PD-(L)1 antibodies. The authors intend to obtain general implications for immunotherapy treatments but in practice their model is used, if I understood it properly, to describe the response to ICI treatments. Immunotherapies are much more than ICI and include cell therapies, i.e. cancer vaccines, oncolytic viruses, CAR-T cells, NK-cells, bispecific antibodies and more. Many of these treatments have been described with mathematical models that are substantially different from the one proposed in this manuscript. For example the response patterns, and the corresponding mathematical models, to CAR-T cell treatments in leukaemias have nothing to do with the "delayed" effect described in this paper for ICI. Thus, the model does not contain essentially new elements, the idea has been previously exploited by other authors and the implications may be restricted to specific immunotherapies. This does not mean that the manuscript is not interesting but that, when revised thoroughly, it would suit more a different type journal such as Scientific Reports, PLOS Computational Biology or Journal of Theoretical Biology.

Beyond that, there are other points that I recommend the authors to revise:

- 1.- The title should focus on the types of immunotherapy used in the manuscript.
- 2.- The exponent $4/5$ in Eq. (1) does not seem natural to me. Different exponents have been used in the literature such as the $3/4$ (the West-Brown-Enquist universal growth law, doi: doi.org/10.1038/35098076) that has been reported to describe tumor growth in-vitro and in some animal models (see e.g. doi:10.1016/S0022-5193(03)00221-2) . An exponent 1 is the classical exponential growth model. Exponents larger than one have been found in untreated human tumors as a result of evolutionary dynamics within heterogeneous tumors (doi: 10.1038/s41567-020-0978-6). But why $4/5$?
- 3.- There are many models of tumor-immune dynamics. It would be convenient to place the model with the state of the art in the field.
- 4.- Tumor growth rate decline is not explained mathematically (or written explicitly in the model equations). Why? The explanation in the text is not precise enough. Moreover at that stage it is not clear why tumor growth rate should decline at all (see again comment 2 for untreated tumors), thus it seems to me that tumor growth rate decline is an effective way of accounting for the effect of treatments.
- 5.- The idea of taking a publicly available dataset of patients with advanced lung cancer from the North Central Cancer Treatment Group (NCCTG) and regarding the survival times of these patients as if they were untreated is not realistic since those patients would probably be heavily treated.
- 6.- Some references supporting the choices for the size at which common malignancies are diagnosed and the lethal tumor burden are necessary.

7.- The authors claim that immune checkpoint inhibitors increase the killing rate of cytotoxic T cells by a multiplication factor: 0-7. I guess they want to mean 1-7. But why 1-7? Please provide a reference.

8.- The fact that the model is able to describe qualitatively the results of the trial (mean and risks) could be a result of the very many parameters present in the model equations (plus the growth rate decay parameters) and not a reflection of the model ability to capture the tumor's biology. Would other (e.g. simpler) models available in the literature fit also the data? I have seen often how a biologically wrong model fits a dataset just because of it having a sufficiently high number of degrees of freedom. Would different models lead to similar results?

Reviewer #2 (Remarks to the Author):

Creemers and colleagues present a very interesting mathematical oncology manuscript that simulates cancer immunotherapy trials with the goal to inform clinical trial design. The article is well written and of high interest and importance to the fields of cancer biology, immunology, clinical oncology, and mathematical biology. The topic is well defined and the results intriguing. To fully evaluate the potential and impact of the work, I have several questions or concerns that should be addressed.

The model without treatment is calibrated to advanced lung cancer treatment NCCTG data as if those were untreated patients (p6, l180f). This makes the model calibration incorrect.

The mathematical model is based on a previous publication (Ref. 27). In that work, the model is not calibrated and validated to any time-resolved data to evaluate dynamics and define predictive power. Of particular interest is the power of 4/5 in the growth term, which is explained in the original work as large parts of the tumor being necrotic. However, in the interaction terms, the variable T is not scaled to account for necrosis, which may severely overestimate the impact of the interactions.

The functional form of the effect of the different therapies is not included, and how the parameters were fitted and what values they have needs to be discussed.

The default values or ranges of all parameters need to be justified. Activated Tcell production rate of $p_s=1$ cell per day and migration rate of $m_s=1$ cell per day sound very low. The introduction of $\Delta\rho$ and ρ_{Δ} are unorthodox to simulate a deceleration of tumor growth. This needs more motivation, especially with the exponent in the growth term already accounting for deceleration.

Immune checkpoint inhibitors increase the killing rate by a multiplication factor of 0-7. Should this be 1-7? How were ranges 0-7 and 0-1 for chemotherapy chosen ?

It is unclear what eqn. 2 is in the model. It is not discussed, and only serves as a sink of activated T cells. Can the exponential decay not be added to Eq. 3 with similar effect?

Large parts of the results should be methods, or are repeats from introduction and methods and should be removed.

P11 l297. what are "reasonably corresponding risk tables". Are there statistical measurements to evaluate goodness of fit?

Figure 3 B has different dynamics than figure 2, where curves separate after 4 months. Unclear why this is different here.

Figure 3C+E labels appear to be messed up and do not enable evaluation of the results.

Reviewer #3 (Remarks to the Author):

The manuscript is well written and interesting, one of the clearer and more accessible manuscripts I have read on this area. I have a few questions

1. The authors state "Since a larger tumor mass is generally more immunogenic..."

Do they have any data to support this supposition? If this is not correct how would that affect their model(s). In general more bulky advanced disease is a poor prognostic factor with both immunotherapy and chemotherapy and recently adjuvant IO therapy (in the absence of any visible tumor) has shown efficacy across several solid tumors in phase 3 trials.

2. They state "As an illustrative example, we took a publicly available dataset of patients with advanced lung cancer from the North Central Cancer Treatment Group (NCCTG) and regarded the survival times of these patients as if they were untreated."

– I am not familiar with this specific cohort however it is doubtful whether they had no antineoplastic treatment assuming the cohort is from the last 20 years the majority (perhaps nearly all) would have received chemotherapy, targeted therapy or perhaps even immunotherapy. Given the authors have used this as model fitting cohort how does the fact this cohort was not untreated affect their other analyses?

If a truly untreated cohort is needed perhaps the placebo group in phase 3 trials in the 2nd or 3rd line setting could be used?

3. Can authors explain why 2 year OS was chosen as the endpoint rather than median OS or 5 year survival?

Response to Reviewers

In our point-by-point response below, we use the following conventions:

- *Reviewer comments are typeset in gray italics.*
- Our response is typeset in normal font.
- Descriptions of changes made to the manuscript are typeset in blue, with references to new or revised text in the revised manuscript **typeset in red**.

Reviewer 1

This paper deals with a very interesting topic: the potential use of in-silico studies to guide clinical trial design of immunotherapy treatments. I liked a lot the style of the paper and fully agree with the insightful claims made by the authors in the introduction. I also agree with their conclusion that in silico trials may provide a fast and cheap approach to verify the robustness of biological assumptions underlying immunotherapy trials and help to scrutinize its design.

That said I do not think Nature Communications is the right place for this paper. First of all other versions of the model (and a similar methodology) has been used in previous works by the same authors. Also, in-silico studies have been used recently in the literature to simulate the outcome (and suggest improvements) of treatments with either chemotherapy, tyrosine kinase inhibitors and anti-CTLA-4-antibodies and anti-PD-(L)1 antibodies. The authors intend to obtain general implications for immunotherapy treatments but in practice their model is used, if I understood it properly, to describe the response to ICI treatments. Immunotherapies are much more than ICI and include cell therapies, i.e. cancer vaccines, oncolytic viruses, CAR-T cells, NK-cells, bispecific antibodies and more. Many of these treatments have been described with mathematical models that are substantially different from the one proposed in this manuscript. As an example the response patterns, and the corresponding mathematical models, to CAR-T cell treatments in leukaemias have nothing to do with the "delayed" effect described in this paper for ICI. Thus, the model does not contain essentially new elements, the idea has been previously exploited by other authors and the implications may be restricted to specific immunotherapies. This does not mean that the manuscript is not interesting but that, when revised thoroughly, it would suit more a different type journal such as Scientific Reports, PLOS Computational Biology or Journal of Theoretical Biology.

We appreciate the frank and honest feedback. However, we feel there might have been a misunderstanding about our paper's intent. We agree and are aware that mathematical models have described many different types of immunotherapy treatments over the years; our manuscript used one specific model, which was not particularly novel from a mathematical point of view. Nevertheless, the primary concern of our manuscript is not how to develop mathematical models for specific immunotherapy treatments but how to *use* such models fruitfully for clinical trial design purposes. Therefore, we exploit model-based *in silico* clinical trials to address specific substantive questions that clinical trialists need to answer, such as: which effect size measurements should be used to define the trial endpoint? Which randomization ratio should be used? How many interim analyses, if any, should be performed? We are not aware of previous work that addresses such questions using mechanistic models in the context of cancer immunotherapy, and we are reasonably certain that the clinical trials community and regulatory bodies are not using such models in practice at this time. This is the gap that our manuscript intends to bridge. Indeed, we feel that immunotherapy trial design could benefit enormously from the important work that the Mathematical Oncology community has done, since the approach is independent of the specific type of immunotherapy being modeled; it could be combined with any existing mathematical model as long as the model can generate realistic survival times and treatment effects at realistic levels of heterogeneity.

We did not submit this paper to PLOS Computational Biology or the Journal of Theoretical Biology as its primary intention is to facilitate the adoption of models developed in Mathematical Oncology for the purpose of trial design. We believe an interdisciplinary journal such as *Nature Communications* is a better

fit for this goal, as doctors or trialists are unlikely to read specialty journals in mathematical or computational biology.

We fundamentally changed our approach in response to this comment to strongly emphasize existing contributions by the Mathematical Oncology community. Most importantly, also considering the reviewer's point 8 below, we implemented and analyzed two additional immunotherapy models by other authors (**lines 53–64, lines 381–387, lines 388–393, Table 1, Table 2, Figure 3**) - alongside our own; indeed, we now feel this should be the default approach since it is crucial to understand how strongly trial design decisions depend on modeling assumptions. We hope these changes will make it evident that this paper is not about one specific mathematical model of one specific type of immunotherapy but about harnessing such models for the purpose of clinical trial design.

1.- The title should focus on the types of immunotherapy used in the manuscript.

As explained above, this manuscript introduces a general approach to trial design that should apply to all types of immunotherapy. Since the types of immunotherapy used in the manuscript are merely examples to illustrate the approach of exploiting *in silico trials* in the trial design process, we deliberately chose to refrain our focus from the different treatments to prevent distracting the readership with a therapy-oriented view. However, we agree that the impression that our approach is restricted to one specific model or one specific type of immunotherapy should be avoided.

We have implemented a further mathematical model by Bekker *et al.* [1] (**lines 53–64, lines 388–393**), which according to the authors, can be used to model the effects of cytotoxic treatments, cell-based immunotherapies, and ICI.

2.- The exponent 4/5 in Eq. (1) does not seem natural to me. Different exponents have been used in the literature such as the 3/4 (the West-Brown-Enquist universal growth law, doi: doi.org/10.1038/35098076) that has been reported to describe tumor growth in-vitro and in some animal models (see e.g. doi:10.1016/S0022-5193(03)00221-2). An exponent 1 is the classical exponential growth model. Exponents larger than one have been found in untreated human tumors as a result of evolutionary dynamics within heterogeneous tumors (doi: 10.1038/s41567-020-0978-6). But why 4/5?

The same issue has been raised by reviewer 2. We based this exponent on the work of Murphy *et al.* [2] (exponent 0.785; source: cell lines). Since the model is merely used as an example to illustrate its application in a novel methodology to support clinical trial design and minor deviations from the exponent do not determine the outcomes of our experiments, we agree that a more commonly used exponent would be more suitable to prevent distracting the readership of the main message.

We changed this number in our model M1 to the 3/4 exponent of the West-Brown-Enquist universal growth law [3] and re-did all analyses with this exponent (**Figure 2, Figure 3, Figure 4, Figure 5, Figure 6, Figure 7**). We also implemented two other models with different growth laws, which did not affect the paper's general conclusions.

3.- There are many models of tumor-immune dynamics. It would be convenient to place the model with the state of the art in the field.

We have added a paragraph to the discussion section emphasizing more clearly the rich literature on mathematical models of tumor-immune dynamics, including references to several reviews (**lines 279–288**).

4.- Tumor growth rate decline is not explained mathematically (or written explicitly in the model equations). Why? The explanation in the text is not precise enough. Moreover at that stage it is not clear why tumor growth rate should decline at all (see again comment 2 for untreated tumors), thus it seems to me that tumor growth rate decline is an effective way of accounting for the effect of treatments.

We used this growth rate decline to increase heterogeneity in our predicted survival times. It felt like a natural assumption that the growth rate does not remain constant over time, but we agree with the criticism that this might implement a treatment effect by proxy.

We removed these parameters from our model (now model M1), which did not affect the conclusions of our paper.

5.- *The idea of taking a publicly available dataset of patients with advanced lung cancer from the North Central Cancer Treatment Group (NCCTG) and regarding the survival times of these patients as if they were untreated is not realistic since those patients would probably be heavily treated.*

We agree that this was confusing; two other reviewers have raised the same issue.

We have addressed it by including a chemotherapy treatment effect in our fit of this cohort (**lines 113–122**), and we have switched to using the more recent CA184-024 trial data as a basis for our predictions (**lines 152–154**). This allows us to estimate what the corresponding untreated cohort would look like by setting the treatment effect back to 0 (placebo arms in **Figure 4, Supplementary Figure S2, Supplementary Figure S3, and Supplementary Figure S4**). Despite all these changes, the paper's general conclusions remained the same.

6.- *Some references supporting the choices for the size at which common malignancies are diagnosed and the lethal tumor burden are necessary.*

We have added references to the size of a tumor at diagnosis (**line 428**). The lethal tumor burden initially used in our model was a simple approximation; this varies widely between patients. Therefore, we now treat the lethal tumor burden as a random variable that spans across one order of magnitude (**lines 426–432**). Our conclusions are unaffected by this change.

7.- *The authors claim that immune checkpoint inhibitors increase the killing rate of cytotoxic T cells by a multiplication factor: 0-7. I guess they want to mean 1-7. But why 1-7? Please provide a reference.*

The same issue has been raised by reviewer 2. The given range for the killing rate increase was an error; this had been taken from the previous paper [4], where we had used the range 1-7 (indeed, not 0-7) in our experiments. In this paper, we would also encourage readers to consider values outside of this range (as long as they are not unphysiologically high). Likewise, the range 0-1 is simply the range of all possible growth rate declines.

In the new version, we no longer give ranges for these parameters (**Table 1, Table 2**), as they are determined by fitting to the CA184-024 cohort (**lines 113–122**), and the ranges have also been affected by our changing of the growth law from exponent 0.8 to 0.75 (see previous points).

8.- *The fact that the model is able to describe qualitatively the results of the trial (mean and risks) could be a result of the very many parameters present in the model equations (plus the growth rate decay parameters) and not a reflection of the model ability to capture the tumor's biology. Would other (e.g. simpler) models available in the literature fit also the data? I have seen often how a biologically wrong model fits a dataset just because of it having a sufficiently high number of degrees of freedom. Would different models lead to similar results?*

It is, of course, true that wrong models can still fit data, especially if they have many parameters. In our case, we do not fit that many parameters to the data shown in Figure 3; all model parameters except for the tumor growth rate and the treatment effect sizes are fixed to their biologically motivated values given in the previous paper and were not fitted to the trial data. This means that the number of parameters fitted ranges from 3-5, depending on the dataset. However, we were intrigued by the reviewer's question about whether other models would lead to similar results.

As mentioned in response to point 1 above, we have implemented two additional models of tumor-immune dynamics [5, 1] (**lines 53–64, lines 381–387, lines 388–393**). Although there are differences in how well these models fit existing trial data despite the same number of parameters being fitted (**Figure 3, Figure S1**), the agreement between the models is remarkable when predicting the aspects of the survival curves most relevant for trial design (**lines 209–216, Figure 5, Supplementary Figure S5**). Furthermore, these new analyses illustrate that the model predictions relevant to trial design do not necessarily depend strongly on model structure, number of parameters, or quality of initial fit.

Reviewer 2

Creemers and colleagues present a very interesting mathematical oncology manuscript that simulates cancer immunotherapy trials with the goal to inform clinical trial design. The article is well written and of high interest and importance to the fields of cancer biology, immunology, clinical oncology, and mathematical biology. The topic is well defined and the results intriguing. To fully evaluate the potential and impact of the work, I have several questions or concerns that should be addressed.

The model without treatment is calibrated to advanced lung cancer treatment NCCTG data as if those were untreated patients (p6, 1180f). This makes the model calibration incorrect.

We agree that this was confusing; two other reviewers have raised the same issue.

We have addressed it by including a chemotherapy treatment effect in our fit of this cohort (**lines 113–122**), and we have switched to using the more recent CA184-024 trial data as a basis for our predictions (**lines 152–154**). This allows us to estimate what the corresponding untreated cohort would look like by setting the treatment effect back to 0 (placebo arms in **Figure 4, Supplementary Figure S2, Supplementary Figure S3, and Supplementary Figure S4**). Despite all these changes, the paper's general conclusions remained the same.

The mathematical model is based on a previous publication (Ref. 27). In that work, the model is not calibrated and validated to any time-resolved data to evaluate dynamics and define predictive power. Of particular interest is the power of 4/5 in the growth term, which is explained in the original work as large parts of the tumor being necrotic. However, in the interaction terms, the variable T is not scaled to account for necrosis, which may severely overestimate the impact of the interactions.

We do not intend to claim that the model(s) accurately predict(s) specific treatments coming from time-resolved data. The time-resolved kinetics will likely be very different across clinical immunotherapy trials owing to differences in patient populations, the standard of care, inclusion criteria, and, most importantly, the specific cancer being treated. Instead, the model(s) aim to generate a (wide) range of plausible trial outcome scenarios to scrutinize the trial design for potential issues. Indeed, given that a wide range of exponents is used to model tumor growth [2], it would be good if the *in silico* trial design would scrutinize several of these.

We changed the exponent to 3/4 and re-did all analyses to illustrate that these choices do not affect our conclusions. See also the related answer to reviewer 1. The impact of interactions is also controlled by the parameter ξ . We use lower values of ξ than other models that do not work with slowing tumor growth; we now discuss this (**lines 408–411, Table 2**). We think this has also become clearer by including the other two models and showing that the impact of the immune system is comparable (**Figure 2**).

The functional form of the effect of the different therapies is not included, and how the parameters were fitted and what values they have needs to be discussed.

We have explicitly specified that therapies are implemented by changing a model parameter to a different value for a set amount of time (**lines 433–440**). In our new version, we use a more straightforward fitting method, Approximate Bayesian Computation (**lines 455–469**), and we make clear that we fit only a small subset of the parameters to the data (**lines 113–122, Table 3**).

The default values or ranges of all parameters need to be justified. Activated T cell production rate of $p_S = 1$ cell per day and migration rate of $m_S = 1$ cell per day sound very low.

We thank the reviewer for checking this so carefully. Unfortunately, the units of several of our parameters in Table 1 were incorrect, which may have led to a misunderstanding. The value $p_S = 1$ means that each T cell takes 1 day for one round of proliferation and 1 day to move from the lymph node to the target site; these values match current immunological knowledge.

We have corrected the units (**Table 1**). In addition, we added a new paragraph to the results section that describes in much more detail how the parameters of each model were set (**lines 398–419**). The value of m_S in M1 is also discussed in **lines 378–380**.

The introduction of $\Delta\rho$ and ρ_δ are unorthodox to simulate a deceleration of tumor growth. This needs more motivation, especially with the exponent in the growth term already accounting for deceleration.

We have removed these parameters. We now keep ρ constant throughout, except when it is affected by therapy. See also related points by reviewer 1.

Immune checkpoint inhibitors increase the killing rate by a multiplication factor of 0-7. Should this be 1-7? How were ranges 0-7 and 0-1 for chemotherapy chosen?

The same issue has been raised by reviewer 1. The given range for the killing rate increase was an error; this had been taken from the previous paper [4], where we had used the range 1-7 (indeed, not 0-7) in our experiments. In this paper, we would also encourage readers to consider values outside of this range (as long as they are not unphysiologically high). Likewise, the range 0-1 is simply the range of all possible growth rate declines.

In the new version, we no longer give ranges for these parameters (**Table 1, Table 2**), as they are determined by fitting to the CA184-024 cohort (**lines 113–122**), and the ranges have also been affected by our changing of the growth law from exponent 0.8 to 0.75 (see previous points).

It is unclear what eqn. 2 is in the model. It is not discussed, and only serves as a sink of activated T cells. Can the exponential decay not be added to Eq. 3 with similar effect?

Our model distinguishes between T cells in the lymph node (N, S) and T cells at the tumor site (I) based on our understanding of antitumoral T cell immune responses [4]. This leads to a delay effect where activated T cells travel to the tumor site before exerting their effect. Indeed, this could probably be simplified at the current parameterization while giving similar predictions. The other two models implemented in the revised manuscript do not make this distinction.

In the revised manuscript, we now explain the difference between the I and N/S populations of T cells and mention that the model could be simplified if desired (see **lines 352–446**, especially **lines 378–380**).

Large parts of the results should be methods, or are repeats from introduction and methods and should be removed.

Although we are unsure about which parts of the results the reviewer was referring to, we suspect they might refer to the initial part of the results section, where we explained our model.

This part and the Methods section have been rewritten in the revised version (see especially **lines 53–64** and **lines 352–446**). Our general goal was to provide an easily accessible description of each model in the main manuscript text, with a more formal mathematical definition in the Methods. We feel that this is justified, given our use of multiple models in the manuscript. We have also added a new Figure showing simulation results from each model (**Figure 2**).

P11 1297. what are “reasonably corresponding risk tables”. Are there statistical measurements to evaluate goodness of fit?

We changed how we fit the models to the data in the revision. We now use Approximate Bayesian Computation (ABC), with the root mean square difference between survival curves as a measurement of fit (**lines 455–469**). We are not comparing the risk tables statistically. Therefore, we have removed this statement from the revision.

Figure 3 B has different dynamics than figure 2, where curves separate after 4 months. Unclear why this is different here.

Figure 2 shows well-fitting runs of each model to each specific dataset, as generated during ABC (see the response to the previous point), whereas Figure 3 (now Figure 4 in the new version) shows prediction based on these fits on new types of immunotherapy.

We have recreated **Figure 4** and explain the difference to **Figure 3** more explicitly and hopefully more clearly (**lines 152–154, lines 455–469**).

Figure 3C+E labels appear to be messed up and do not enable evaluation of the results.

We fixed this issue in the new version (**Figure 4**).

Reviewer 3

The manuscript is well written and interesting, one of the clearer and more accessible manuscripts I have read on this area. I have a few questions.

1. The authors state "Since a larger tumor mass is generally more immunogenic..." Do they have any data to support this supposition? If this is not correct how would that affect their model(s). In general more bulky advanced disease is a poor prognostic factor with both immunotherapy and chemotherapy and recently adjuvant IO therapy (in the absence of any visible tumor) has shown efficacy across several solid tumors in phase 3 trials.

We agree that this was poorly phrased. Indeed, in our previous work [4], we have argued that tumor size (or growth rate) on its own is a poor prognostic factor since the ultimate immunogenicity also depends on many other parameters. In our model, small tumors (say, of approximately 10^5 cells) lead to more antigen presentation than microscopic tumors (say, containing only a dozen cells). As the tumor grows, antigen presentation saturates. The immunogenicity itself is controlled by other model parameters (such as the killing rate) because we do not think that tumor size alone dictates the immune response.

We have rephrased this sentence to be more precise and removed the reference to immunogenicity (**lines 357–360**).

2. They state "As an illustrative example, we took a publicly available dataset of patients with advanced lung cancer from the North Central Cancer Treatment Group (NCCTG) and regarded the survival times of these patients as if they were untreated." – I am not familiar with this specific cohort however it is doubtful whether they had no antineoplastic treatment assuming the cohort is from the last 20 years the majority (perhaps nearly all) would have received chemotherapy, targeted therapy or perhaps even immunotherapy. Given the authors have used this as model fitting cohort how does the fact this cohort was not untreated affect their other analyses? If a truly untreated cohort is needed perhaps the placebo group in phase 3 trials in the 2nd or 3rd line setting could be used?

Since our model already fitted this cohort without assuming a treatment effect, we had regarded it as untreated in the paper to avoid complicating things. We regret the confusion this has caused; see also related comments by other reviewers.

We have now included a chemotherapy treatment effect in our fit of this cohort (**lines 113–122**), and have switched to using the more recent CA184-024 data as a basis for our predictions (**lines 152–154**). This allows us to estimate what the corresponding untreated cohort would look like by setting the treatment effect back to 0 (placebo arms in **Figure 4, Supplementary Figure S2, Supplementary Figure S3, and Supplementary Figure S4**). Despite all these changes, the general conclusions of the paper remained the same.

Can authors explain why 2 year OS was chosen as the endpoint rather than median OS or 5 year survival?

We acknowledge that the rationale for the endpoint selection was lacking in our manuscript. In general, Overall Survival is regarded as the ‘gold standard’ primary clinical endpoint in oncology trials [6]. The 2-year OS was selected as a primary endpoint for two reasons. First, in contrast to the median OS, the 2-year OS is fixed in time, making the endpoint independent of the survival rate and more convenient for trial planning. Second, while long-term overall survival rates are insightful, a 5-year overall survival endpoint would be too long to serve as a primary endpoint from an ethical and marketing perspective. Therefore, we selected the 2-year OS endpoint to meet the robust 5-year overall survival and surrogate endpoints as the progression-free survival or time-to-progression in the middle.

We added a brief explanation of this choice with some motivating references (**lines 50–52**).

Reviewer 4

This manuscript states that the immunotherapy-derived survival patterns, such as delayed curve separation and plateauing curve of the treatment arm, arise naturally due to the biological interactions in the tumor

environment and thus proposes the in silico trial design strategy, which is capable of translating these biological interactions into survival kinetics, to improve the cancer immunotherapy trial designs. Although this work discusses an important question of interest and develops a potentially useful tool for trial designs provided that the model assumptions can be validated, it has some critical concerns that need to be dressed.

1. The authors claim that an in silico immunotherapy trial is based on clear-cut biological assumptions and provides an intuitive means to predict risk profiles and treatment efficacy. Hence, the in silico trials could equip researchers with a tool to verify trial designs and analysis strategies of upcoming trials before the trials execution. Under this context, the validity of the biological assumptions used to build the ODE model and the choice of the model parameters such as the tumor growth rate, growth rate decline and decline decay rate, becomes critical to the performance of the in silico trial. Although the authors also show that the simulations replicate late-stage immunotherapy or combination trials realistically and capture their typical survival kinetics, the verification is only based on a limited number of case studies, and is difficult to capture a wide spectrum of the realistic scenarios that may arise.

We thank the reviewer for this vital comment, which prompted us to rethink how we framed our contribution. We did not intend to claim that *in silico* trials would be able to accurately predict, at a quantitative level, each individual patient's response to a novel treatment. We instead wanted to show how *in silico* trials could be used to generate principled predictions about a range of *possible* outcomes that could be expected. Indeed, we would argue that investigators should base their choices on the predictions of multiple models – there are many choices available in the Mathematical Oncology literature – that may differ from each other. Such differences arise due to choices in modeling assumptions and parameter uncertainty, which will be inevitable. Nevertheless, we feel that this way of embracing structural uncertainty and working with it is better than expecting a single model to perfectly predict the results of novel trials – if such a model were available, we would perhaps not need to do the trial at all.

In our revised manuscript, we use three different models instead of just one (**Figure 2, lines 53–64**) and show explicitly that they can generate different quantitative predictions (**Figure 4, Supplementary Figure S2, Supplementary Figure S3, and Supplementary Figure S4**). We also show that the models often agree on essential qualitative aspects of the predicted survival curves despite those differences. For example, all three models predict a delayed curve separation in a chemoimmunotherapy vs. immunotherapy trial (**lines 174–178, Figure 4, Supplementary Figure S3, and Supplementary Figure S4**). We have also carefully revised the text to rephrase statements that could be interpreted as claims that we can accurately predict the results of a novel trial on a quantitative level.

a. Page 10, Lines 281-285, the authors illustrate the use of the in silico approach by fitting the simulation to three different datasets. Can the authors use one dataset to tune the model parameters and then use the other two datasets to validate the goodness of the fit of the in silico model? Or can the authors tune the model parameters based on the three datasets and validate its prediction power by applying the model in other cancer immunotherapy studies already completed? By doing so, we can avoid the risk of model over-fitting and evaluate the a priori prediction power of the in silico model.

We agree that the analysis suggested by the reviewer would be helpful if our goal were to predict the results of clinical trials quantitatively. However, as argued in response to the previous point, this is not our intention. We do not aim to predict survival curves accurately, but we wish to predict essential qualitative features of the curves. Further, we doubt that the baseline patient populations are necessarily comparable between different clinical trials, so we would also need to conduct matching of model parameters on patient covariates to be able to conduct such an analysis.

In our revision, we have fitted three models to the same dataset (**Figure 3, Supplementary Figure S1**) and generated predictions about novel immunotherapies from these fits (**Figure 4, Supplementary Figure S2, Supplementary Figure S3, and Supplementary Figure S4**). This immediately shows that the quantitative predictions differ and are model-specific. However, as we emphasize much more strongly now, the qualitative aspects of the survival curves – the important characteristics of designing the trial – are comparable (**lines 190–216 and especially lines 209–216**). Throughout the entire text, we have also worked to clarify that our goal is not an accurate quantitative prediction of each patient's outcome, which we do not think is realistic for the time being.

2. Page 12-13, Lines 351-360, the authors claim that the flexibility of *in silico* trials lies in their ability to incorporate complex treatment regimens and uses an example to support the claim. In this example, the investigator would be interested in estimating the survival curves and underlying hazards ratio over time under the scenarios of 3D and 3E, where a crossing- survival-curves pattern or a temporary curve separation pattern are manifested. It would be interesting to know, if the investigator uses the *in silico* model to predict the trial outcome and survival patterns, how should they estimate or report the hazards ratio under such complex survival patterns (3D or 3E) after the data is collected? Would the trial report the true hazards ratio used to simulate the *in silico* model or the hazard ratio estimated by the Cox proportional hazards model?

Indeed, the occurrence of crossing survival curves presents a challenge for quantifying the treatment effect. We would argue that in cases where survival curves cross, the hazards ratio should not be used as an effect size, but which effect size would be appropriate is a difficult question and its answer would depend on the specific clinical considerations at hand. To our knowledge, this is an active area of research in the field of trial design, and we consider this to extend beyond the scope of this manuscript. The reviewer's suggestion to fit an *in silico* model to such data and use the estimated treatment effect as the effect size is very interesting, but it has the drawback that its interpretation would be model-dependent, which may not be desirable. However, a related case arises when treatment effects are transient, i.e., when the separation of survival curves is only temporary.

In the revised manuscript, we now explicitly discuss the issue of temporarily separating survival curves and how the hazards ratio can be used to quantify the treatment effect in such a case, even if the proportional hazards assumption is not fully met (lines 190–216, especially lines 199–200).

3. Page 14, Line 273-393, the authors compared the power of two testing procedures, the log-rank test versus Pearson's Chi-square test. I don't think the two testing procedures are comparable, as they assume different primary endpoints; the log-rank test assumes the time-to-event endpoint and uses hazard ratio to measure the treatment effect, whereas the Pearson's Chi-square test assumes the responder rate endpoint and uses the responder rate to quantify the treatment effect. Hence, it is not appropriate to evaluate the strength of different treatment effect measures under different primary endpoint.

We agree that this section was misleading: the power is not directly comparable between the two methods, as they measure different things. As the reviewer writes, the choice of effect size to measure the treatment effect leads to the observed differences in power.

We have revised this section of the results to avoid this misunderstanding and make clear that the comparison is between effect sizes, not between statistical tests (lines 190–216, Figure 5). We hope the revised version clarifies that we intend to illustrate how *in silico* trials can be used to choose an appropriate effect size.

4. Line 166-167, the authors state that "Once a cancer reaches a diagnosis threshold, immune checkpoint inhibitors (ICI) increase the killing rate of cytotoxic T cells". Does the assumption reflect the indirect mechanism of action of ICI, so that the treatment effect would be manifested after a lag period rather than immediately?

This appears to be a misunderstanding. The effect of ICI can only start after diagnosis simply because the patient's treatment only starts after diagnosis. After the start of the treatment, there is a further delay until the treatment effect is manifested in the survival curves because the treatment works only in a subset of the patients; in other words, patients that would die rapidly after diagnosis are unlikely to benefit from the treatment, and so the survival curves do not differ much between the arms at early times.

We have rephrased the sentence and expanded on how we implement treatment in our model (lines 433–440).

Minor comments:

1. Line 134, not all parameters in the equation systems are clearly defined.

In the new manuscript, the Methods section has been completely rewritten (lines 352–446). We defined all parameters clearly and added two new tables explaining the parameters and their default values (Table 1,

Table 2).

2. “hazard ratio” should be revised to “hazards ratio” throughout the manuscript.

We agree that “hazards ratio” would be more correct from a purely grammatical point of view, but since “hazard ratio” is used far more commonly in the literature, we would like to keep this wording to avoid confusion.

3. Line 383, the word “underestimation” should be “overestimation”, as the PHA-dependent methods would lead to an underpowered study when the proportional hazards assumption no longer holds.

This entire section has been rewritten in response to point 3 above (lines 190–216).

References

- [1] Bekker RA, Zahid MU, Binning JM, Spring BQ, Hwu P, Pilon-Thomas S, and Enderling H. Rethinking the immunotherapy numbers game. *Journal for ImmunoTherapy of Cancer*, 10(7):e005107, 2022. doi:10.1136/jitc-2022-005107.
- [2] Murphy H, Jaafari H, and Dobrovolny HM. Differences in predictions of ode models of tumor growth: a cautionary example. *BMC Cancer*, 16(1), 2016. doi:10.1186/s12885-016-2164-x.
- [3] West GB, Brown JH, and Enquist BJ. A general model for ontogenetic growth. *Nature*, 413(6856):628–631, 2001. doi:10.1038/35098076.
- [4] Creemers JHA, Lesterhuis WJ, Mehra N, Gerritsen WR, Figdor CG, de Vries IJM, and Textor J. A tipping point in cancer-immune dynamics leads to divergent immunotherapy responses and hampers biomarker discovery. *Journal for ImmunoTherapy of Cancer*, 9(5):e002032, 2021. doi:10.1136/jitc-2020-002032.
- [5] Tsur N, Kogan Y, Rehm M, and Agur Z. Response of patients with melanoma to immune checkpoint blockade – insights gleaned from analysis of a new mathematical mechanistic model. *Journal of Theoretical Biology*, 485:110033, 2020. doi:10.1016/j.jtbi.2019.110033.
- [6] Delgado A and Guddati AK. Clinical endpoints in oncology - a primer. *Am J Cancer Res*, 11(4):1121–1131, 2021.

REVIEWERS' COMMENTS

Reviewer #1 (Remarks to the Author):

First of all I would like to thank the authors for their revision of the manuscript that has certainly been improved substantially. The changes in the models make them more reasonable and better justified giving more confidence in the results.

However as the authors themselves acknowledge in the introduction, there have been other papers reporting in-silico clinical trials in oncology [56,57], some of them even analyzing specifically immunotherapy treatments [58,59,60]. Aspects such as the optimal treatment schedules or the number of patients to be included to obtain significance between different treatment arms have been previously considered in those manuscripts.

In this paper there are some aspects of clinical trial design that have been addressed in more detail. Thus I still think that this paper is publishable but does not have the level of novelty required to be published at Nature Communications.

Reviewer #2 (Remarks to the Author):

The authors have done an excellent job to revise their manuscript, and I congratulate the team on a very strong study.

Reviewer #3 (Remarks to the Author):

The authors have addressed my comments comprehensively.

Reviewer #4 (Remarks to the Author):

The authors have adequately addressed my review comments.

Response to Reviewers

In our point-by-point response below, we use the following conventions:

- *Reviewer comments are typeset in gray italics.*
- Our response is typeset in normal font.
- Descriptions of changes made to the manuscript are typeset in blue, with references to new or revised text in the revised manuscript **typeset in red**.

In the attached marked-up manuscript file, we have highlighted the key added or revised parts of text that we refer to below.

Reviewer 1

First of all I would like to thank the authors for their revision of the manuscript that has certainly been improved substantially. The changes in the models make them more reasonable and better justified giving more confidence in the results.

However as the authors themselves acknowledge in the introduction, there have been other papers reporting in-silico clinical trials in oncology [56,57], some of them even analyzing specifically immunotherapy treatments [58,59,60]. Aspects such as the optimal treatment schedules or the number of patients to be included to obtain significance between different treatment arms have been previously considered in those manuscripts. In this paper there are some aspects of clinical trial design that have been addressed in more detail. Thus I still think that this paper is publishable but does not have the level of novelty required to be published at Nature Communications.

We looked carefully again at these references cited by ourselves and the reviewer, and we can still confidently state that these papers primarily focus on aspects other than trial design, and do not address the specific issue of immunotherapy-specific response patterns that is the main focus of this manuscript.

We have revised the third paragraph in the Discussion section (page 7) to make the difference between references 56-60 and or work more clear.

Reviewer 2

The authors have done an excellent job to revise their manuscript, and I congratulate the team on a very strong study.

We thank the reviewer for their positive assessment and their helpful comments.

Reviewer 3

The authors have addressed my comments comprehensively.

We thank the reviewer for their positive assessment and their helpful comments.

Reviewer 4

The authors have adequately addressed my review comments.

We thank the reviewer for their positive assessment and their helpful comments.